# Viscoelastic and Electromagnetic Materials with Nonlinear Memory

**DOI:** 10.3390/ma15196804

**Published:** 2022-09-30

**Authors:** Claudio Giorgi, John Murrough Golden

**Affiliations:** 1Dipartimento di Ingegneria Civile, Architettura, Territorio, Ambiente e di Matematica, Università di Brescia, 25133 Brescia, Italy; 2Environmental Sustainability and Health Institute, School of Mathematical Sciences, Technological University Dublin, D07 ADY7 Dublin, Ireland

**Keywords:** materials with memory, nonlinear viscoelasticity, electric conductors with nonlinear memory, free energy, dissipation, energy estimates, 73B30, 73F05, 30E20

## Abstract

A method is presented for generating free energies relating to nonlinear constitutive equations with memory from known free energies associated with hereditary linear theories. Some applications to viscoelastic solids and hereditary electrical conductors are presented. These new free energies are then used to obtain estimates for nonlinear integro-differential evolution problems describing the behavior of nonlinear plasmas with memory.

## 1. Introduction

The theory of materials with memory was designed to provide a wide range of material models, including models of viscoelastic materials, dielectrics and heat conductors with memory. The great majority of them can be characterized by memory functionals: reversible changes are described by the instantaneous response while dissipativity is expressed by the dependence on histories (see [1,2,3,4] and references therein). Usually, the Boltzmann superposition principle was employed to derive a linear constitutive equation for the response of the materials [5]. However, when the behavior of a given material is nonlinear, the Boltzmann superposition principle is not applicable; so a constitutive equation has to be sought by other means.

While more than one approach to nonlinear viscoelasticity was being explored in the literature, it is worth mentioning here the pioneering work of Schapery [6]. By means of a time scale shift factor, he derived single-integral nonlinear viscoelastic constitutive equations from the thermodynamic theory of irreversible processes; these equations are very similar to the Boltzmann superposition integral form and, similar to linear viscoelasticity, still consider the strains to be infinitesimal. A different approach has been employed by Findley et al. [7]. To describe nonlinear memory, they considered a polynomial expansion of a multiple-integral expression so that the dependence of strain on the stress history, and vice versa, is nonlinear even for small deformations. The mathematical complexity of this formulation is too complicated for use in many situations. Another approach involves objective rate equations by means of a thermodynamically consistent scheme which naturally allows the construction of nonlinear viscoelastic models (see [8]). More recently, in [9,10], nonlinear mechanical viscous effects are described by assuming a semi-continuum theoretical model with a geometric nonlinearity.

Although many nonlinear models with memory have been developed, this topic is open to many important improvements. A method is presented here for generating nonlinear constitutive equations from known linear theories characterized by memory functionals. The novelty of our procedure is based on the properties of the free energy functionals.

Memory response functionals lead to the difficulty of determining coherent free energy functionals even if the material response is linear [11,12]. A fundamental property of materials with memory is that there is in general no unique free energy density (we henceforth omit “density”) associated with a given state but rather a convex set of functionals which obey the requirements of a free energy ([13,14] and earlier references therein).

In recent years, explicit formulae have been given for the minimum free energy associated with linear constitutive equations [15,16,17]. The case of fluids is discussed with in [18]. In addition, for relaxation functions given by sums of decaying exponentials (discrete spectrum model) and generalizations of these, explicit expressions have been presented for the maximum and intermediate free energies [19,20,21]. Based on this work, an expression for a more centrally located free energy has been presented [21]. Both the isothermal and non-isothermal cases have been considered [22,23].

If the relaxation function is an integral over decaying exponentials (continuous spectrum model and generalizations), then an explicit expression can be given for the minimum free energy [24]. In this case, the maximum free energy is the work function [21,23,25].

Free energies relating to heat conductors with memory are considered in [26]. There are similarities between such materials, as described in [27], and electrical conductors with memory, which are of interest in the present work.

The concept of equivalent classes of states or minimal states, based on the work of Noll [28], is explored in the context of linear models [29,30,31]. One recent result is that for materials with relaxation functions given by sums of exponentials and generalizations of these, the minimal states are usually non-singleton, while for integrals over exponentials, they are singleton [21,23,25]. A more general point of view on this topic is adopted on page 365 of [1].

In the early sections, free energies with quadratic memory terms—which yield constitutive equations with linear memory terms—are discussed.

A method is presented for generating free energies and nonlinear constitutive equations from known free energies associated with linear theories ([1], p. 112). Examples are discussed for nonlinear viscoelastic theories. Such new free energies are then used to prove a boundedness result for integro-differential equations describing the behavior of nonlinear electromagnetic systems, specifically electrical conductors with memory.

In Section 2, the central result concerning free energies for nonlinear systems is proved on a general vector space—which may be taken as relating to mechanical, thermal or electromagnetic systems or a combination of these. Minimal states are defined and discussed. In Section 3, free energies with quadratic memory terms and higher-order contributions are considered. The issue of whether the quadratic memory terms are positive definite or positive semi-definite is relevant for applications in later sections. This question is explored in Section 4 in the context of minimal states. In Section 5, examples relating to viscoelastic materials are discussed, while in Section 6 and Section 7, electrical conductors with memory are considered, and we obtain estimates for nonlinear integro-differential evolution problems describing the behavior of nonlinear plasmas with memory. While the discussion of these sections is assumed to apply to a small element of the body $B\subset R^{3}$ centered at a point x∈B, we shall omit explicit mention of x. Material deformation, while implicitly included in the general developments of Section 2, are not of primary interest in the present work.

**Notation.** 
*On the matter of notation, vectors are denoted by lowercase and uppercase boldface characters and scalars by ordinary script. The real line is denoted by R, the non-negative reals by $R^{+}$ and the strictly positive reals by $R^{++}$, while the complex plane is C. Let V be a vector space with norm |·|. The dot product V×V↦R will indicate the scalar product in V. The dyadic product is denoted by *⊗*; in indicial notation, ${\left(u⊗v\right)}_{ij}=u_{i}v_{j}$. The space of linear operators V↦V is denoted by Lin(V) and the double dot operation, :, will denote the scalar product in Lin(V). The repeated suffix convention is in operation and used where appropriate.*


## 2. Free Energies for Nonlinear Systems

We consider, in this section, a method for generating free energies relating to systems with nonlinear, memory-dependent constitutive equations. Independent and dependent field variables will be defined on a general vector space V. Let $\Lambda^{t}:R^{+}\mapsto V$ and Λ:R↦V be the history and present value of the independent field variable, where the standard notation
(1)$$\Lambda^{t}\left(s\right)=\Lambda \left(t-s\right),\;\;\;\;s\in R^{+}$$
is understood. The dependent field variable Σ:R↦V is given by a constitutive relation
$$\Sigma \left(t\right)=\tilde{\Sigma }\left(\Lambda^{t},\Lambda \left(t\right)\right).$$

We will assume that Λ(·) belongs to a suitable function space with norm ∥·∥. As noted earlier, the free energy associated with a given state of the material is not in general a uniquely defined quantity. Free energies associated with a given state form a bounded convex set F [14]. Let us denote by $\psi :R\mapsto R^{+}$ a member of F. We have
(2)$$\psi \left(t\right)=\tilde{\psi }\left(\Lambda^{t},\Lambda \left(t\right)\right),$$
where $\tilde{\psi }$ is a nonlinear continuously differentiable function of the present value Λ(t) and a nonlinear functional of the history $\Lambda^{t}$ with a Fréchet differential that is continuous in its arguments. The static or equilibrium history is given by
(3)$$\Lambda^{†}\left(s\right)=\Lambda \left(t\right),\;\;\;\;s\in R^{+}.$$

In particular, $0^{†}$ denotes the zero history,
$$0^{†}\left(s\right)=0,\;\;\;\;s\in R^{+},$$
where 0 is the zero vector in V. Note that the requirement
(4)$$\tilde{\psi }\left(0^{†},0\right)=0$$
is imposed on all free energies. This eliminates the arbitrary constant associated with all physical energies. However, the intrinsic arbitrariness associated with free energies of materials with memory remains (see [32,33,34]).

We define the equilibrium free energy by
(5)$$\phi \left(t\right)=\tilde{\phi }\left(\Lambda \left(t\right)\right):=\tilde{\psi }\left(\Lambda^{†},\Lambda \left(t\right)\right).$$
it is assumed that this is a positive definite function of Λ(t).

The quantity
$$W\left(t\right)=\tilde{W}\left(\Lambda^{t},\Lambda \left(t\right)\right),$$
is the work function, which represents the work completed on the element up to time *t*. It is assumed to be given by
(6)$$W\left(t\right)=\int_{-\infty }^{t}\Sigma \left(s\right)\cdot \dot{\Lambda }\left(s\right)ds,$$
where the superimposed dot represents time differentiation and the centered dot represents the scalar product in V. The convergence of the integral in relation (Equation 6) imposes restrictions on the behavior of Λ in the distant past. The work completed over the time interval $\left[t_{1},t_{2}\right]$ is given by
(7)$$W\left(t_{1},t_{2}\right)=\int_{t_{1}}^{t_{2}}\Sigma \left(s\right)\cdot \dot{\Lambda }\left(s\right)ds.$$

Based on consequences of the second law of thermodynamics derived in [32,35], in a continuum mechanics context (and easily generalizable to other dissipitive systems, including electromagnetism [36]), we assign the following (defining) properties to a free energy [14,32,35]:

**Property** **1.**

(8)
$$\frac{\partial \psi (t)}{\partial \Lambda (t)}=\Sigma \left(t\right),$$

*where Σ:R↦V is the dependent field variable.*


**Property** **2.**
*For any history $\Lambda^{t}$ and present value Λ(t)*

(9)
$$\tilde{\psi }\left(\Lambda^{t},\Lambda \left(t\right)\right)\geq \tilde{\phi }\left(\Lambda \left(t\right)\right).$$



**Property** **3.**
*Finally, assuming that *Λ* is continuously differentiable,*

(10)
$$\dot{\psi }\left(t\right)\leq \dot{W}\left(t\right)=\Sigma \left(t\right)\cdot \dot{\Lambda }\left(t\right).$$



These are referred to as the Graffi conditions for a free energy. The relationship between them and the Coleman and Owen [37] definition of a free energy are explored, for a linear theory, by Del Piero and Deseri [30,31]; see also [15].

Note that equality occurs by definition in Property 2 for the static history. Together with the condition (Equation 4), Property 2 implies that all free energies are non-negative-valued functions.

**Definition** **1.**
*We say that a non-negative-valued function ψ≥0 is positive definite if it is zero only for the zero history, $\Lambda^{t}=0^{†}$, and the current zero value Λ(t)=0; otherwise, it is positive semi-definite.*


We can write (Equation 10) as an equality in the form
(11)$$\dot{\psi }\left(t\right)+D\left(t\right)=\dot{W}\left(t\right),$$
where $D\left(t\right):=\dot{W}\left(t\right)-\dot{\psi }\left(t\right)\geq 0$ is the rate of dissipation.

The work function has Properties 1–3 of a free energy, though for zero rate of dissipation. However, it does not have a fourth property, as discussed on page 435 of [1], which applies to all other known free energies. This relates to the fact that for a long established periodic history, we must have ψ(t+T)=ψ(t), where *T* is the period of the history. This may not crucially affect the usage of W(t) as a free energy in all circumstances, but it should be considered with caution.

Let us now state and prove the central result of this section. Let $\psi_{1}\left(t\right)$, $\psi_{2}\left(t\right)$, $\ldots ,\psi_{n}\left(t\right)$ be a set of *n* free energies relating to a state $\left(\Lambda^{t},\Lambda \left(t\right)\right)$ in a given material or perhaps in different materials at time *t*. To allow for the latter possibility, we assign to each $\psi_{i}\left(t\right),\;i=1,2,\ldots n$, different constitutive equations
$$\Sigma_{i}\left(t\right)={\tilde{\Sigma }}_{i}\left(\Lambda^{t},\Lambda \left(t\right)\right),$$
and work functions $W_{i}\left(t\right)$, where
$$W_{i}\left(t\right)=\int_{-\infty }^{t}\Sigma_{i}\left(s\right)\cdot \dot{\Lambda }\left(s\right)ds.$$

Accordingly,
(12)$$\frac{\partial \psi_{i}\left(t\right)}{\partial \Lambda (t)}=\Sigma_{i}\left(t\right),\;\;\;\;i=1,2,\ldots n$$
and
$${\dot{\psi }}_{i}\left(t\right)\leq {\dot{W}}_{i}\left(\Lambda^{t},\Lambda \left(t\right)\right)=\Sigma_{i}\left(t\right)\cdot \dot{\Lambda }\left(t\right),\;\;\;\;i=1,2,\ldots n.$$

If all the free energies belong to the same material, the dependent field variables $\Sigma_{i}$ are all equal, and the index *i* refers to different free energies of the same material.

**Proposition** **1.**
*The quantity*

(13)
$$\psi \left(t\right)=f(\psi_{1}\left(t\right),\psi_{2}\left(t\right),\ldots \psi_{n}\left(t\right))$$


*is a free energy for the state $\left(\Lambda^{t},\Lambda \left(t\right)\right)$ with the dependent field given by*

(14)
$$\Sigma \left(t\right)=\sum_{i=1}^{n}\frac{\partial f}{\partial \psi_{i}\left(t\right)}\Sigma_{i}\left(t\right),$$


*provided that $f:{\left(R^{+}\right)}^{n}\mapsto R^{+}$ has the properties*

(15)
$$\frac{\partial f}{\partial y_{i}}\left(y_{1},y_{2},\ldots ,y_{n}\right)\geq 0,\;\;\;\;i=1,2,\ldots n,$$


*and*

(16)
f(0,0,…0)=0.



**Proof.** We have
$$\begin{array}{c} \dot{\psi }\left(t\right)=\sum_{i=1}^{n}\frac{\partial f}{\partial \psi_{i}\left(t\right)}{\dot{\psi }}_{i}\left(t\right)\leq \sum_{i=1}^{n}\frac{\partial f}{\partial \psi_{i}\left(t\right)}\Sigma_{i}\left(t\right)\cdot \dot{\Lambda }\left(t\right)=\Sigma \left(t\right)\cdot \dot{\Lambda }\left(t\right) \end{array}$$
where Σ is defined by (Equation 14). Moreover, from (Equation 11), we obtain the rate of dissipation
(17)$$D\left(t\right)=\sum_{i=1}^{n}\frac{\partial f}{\partial \psi_{i}\left(t\right)}D_{i}\left(t\right)\geq 0.$$These relations essentially state Property 3. In addition, by virtue of (Equation 12) and (Equation 14),
(18)$$\frac{\partial \psi (t)}{\partial \Lambda (t)}=\Sigma \left(t\right),$$
which is Property 1. It follows from (Equation 13), by taking the stationary limit of the history $\Lambda^{t}$, that
(19)$$\phi \left(t\right)=f\left(\phi_{1}\left(t\right),\phi_{2}\left(t\right),\ldots \phi_{n}\left(t\right)\right)\leq f\left(\psi_{1}\left(t\right),\psi_{2}\left(t\right),\ldots \psi_{n}\left(t\right)\right)=\psi \left(t\right),$$
which is Property 2. Finally, by virtue of (Equation 16), ψ satisfies the normalization condition (Equation 4) since every $\psi_{i}$, i=1,2,…n has this property. □

This result allows us to build free energies and constitutive dependent fields relating to nonlinear systems from those associated with basic constitutive equations with linear memory terms (for which many explicit forms exist [15,16,17,18,19,20,21,22,38]) though, in fact, the $\psi_{i},\;\;i=1,2,\ldots n$ may be any choice of free energies. Specifically, the result can be used as follows: assume we have a nonlinear dependent field variable of the form (Equation 14), where *f* obeys (Equation 15) and $\Sigma_{i}$ is determined by (Equation 12). Then, (Equation 13) immediately yields a free energy with a rate of dissipation given by (Equation 17) and dependent variable Σ(t) generated through (Equation 18). This is the way it is used in Section 5.1 and Section 6.1.

Taking *f* to be an analytic function of its arguments at the origin, we can write
(20)$$\psi \left(t\right)=\sum_{i=1}^{n}\lambda_{i}\psi_{i}\left(t\right)+\sum_{j,k=1}^{n}\mu_{jk}\psi_{j}\left(t\right)\psi_{k}\left(t\right)+\;\mathrm{higher}\;\mathrm{powers}.$$

A constant term is excluded by (Equation 4) and (Equation 16). If we omit higher powers, assumption (Equation 15) takes the form
$$\lambda_{i}+\sum_{k=1}^{n}\left(\mu_{ik}+\mu_{ki}\right)\psi_{k}\left(t\right)\geq 0,\;i=1,2,\ldots n.$$

In particular, taking ψ to be a linear combination of the $\psi_{i}$, it follows that
(21)$$\lambda_{i}\geq 0,\;\;\;\;i=1,2,\ldots n.$$

If we are dealing with free energies relating to the same linear material with dependent field $\Sigma_{\ell }$, then $\Sigma_{i}\left(t\right)=\Sigma_{\ell }\left(t\right)$, i=1,…,n, and (Equation 14) becomes
$$\Sigma \left(t\right)=\kappa \left(t\right)\Sigma_{\ell }\left(t\right),\;\;\;\;\;\;\;\;\kappa \left(t\right)=\sum_{i=1}^{n}\frac{\partial f}{\partial \psi_{i}\left(t\right)}.$$

In the linear case, $\Sigma \left(t\right)=\Sigma_{\ell }\left(t\right)$ and then κ(t)=1. When higher powers are neglected in (Equation 20), this gives
(22)$$\sum_{i=1}^{n}\lambda_{i}:=\left(\sum_{i=1}^{n}\frac{\partial f}{\partial \psi_{i}\left(t\right)}\right){|}_{\psi_{1}=\psi_{2}=\cdots =0}=1$$
which, together with (Equation 21), amounts to convexity.

**Remark** **1.**
*Relations (Equation 21) and (Equation 22) must hold in the general case, with higher powers included, if states exist for which the higher powers are negligible.*


### 2.1. Relative Histories

We can write (Equation 2) in the form
(23)$$\psi \left(t\right)=\hat{\psi }\left(\Lambda_{r}^{t},\Lambda \left(t\right)\right),$$
where $\Lambda_{r}^{t}$ is the *relative history* defined by
(24)$$\Lambda_{r}^{t}\left(s\right)=\Lambda^{t}\left(s\right)-\Lambda \left(t\right).$$

Property 1, given by (Equation 8), becomes
(25)$$\frac{\partial }{\partial \Lambda (t)}\hat{\psi }\left(\Lambda_{r}^{t},\Lambda \left(t\right)\right)-D_{\Lambda }\hat{\psi }\left(\Lambda_{r}^{t},\Lambda \left(t\right)\right)=\hat{\Sigma }\left(\Lambda_{r}^{t},\Lambda \left(t\right)\right)=\Sigma \left(t\right),$$
where the second term on the left is defined by the requirement that, for any Λ(t),
(26)$$D_{\Lambda }\hat{\psi }\left(\Lambda_{r}^{t},\Lambda \left(t\right)\right)\cdot \Lambda \left(t\right)=\delta \hat{\psi }\left(\Lambda_{r}^{t},\Lambda \left(t\right)|\Lambda^{†}\right).$$

The quantity $\delta \hat{\psi }$ is the Fréchet differential of $\hat{\psi }$ at $\Lambda_{r}^{t}$ in the direction $\Lambda^{†}$ which denotes a static history given by (Equation 3), for arbitrary Λ(t). We put
(27)$$\hat{\psi }\left(\Lambda_{r}^{t},\Lambda \left(t\right)\right)=\tilde{\phi }\left(\Lambda \left(t\right)\right)+{\hat{\psi }}_{r}\left(\Lambda_{r}^{t},\Lambda \left(t\right)\right)=\phi \left(t\right)+\psi_{r}\left(t\right)$$
where $\tilde{\phi }$, defined by (Equation 5), is given here by
(28)$$\tilde{\phi }\left(\Lambda \left(t\right)\right)=\hat{\psi }\left(0^{†},\Lambda \left(t\right)\right)$$
and ${\hat{\psi }}_{r}=\hat{\psi }-\tilde{\phi }\geq 0$ represents the history-dependent part of the free energy.

### 2.2. Summed Histories

It is sometimes the case, as in one of the physical configurations dealt with in later sections (and in [26]), that the work function has the form
(29)$$W\left(t\right)=\int_{-\infty }^{t}\Sigma \left(s\right)\cdot \Lambda \left(s\right)ds,$$
with no time derivative on the independent field. Formally, we transform this into (Equation 6) as follows: define the summed past history by
(30)$$\hat{\Lambda }\left(t\right)=\int_{0}^{t}\Lambda \left(u\right)du=\int_{0}^{t}\Lambda^{t}\left(s\right)ds,\;\;\;\;\;\;\;\;{\hat{\Lambda }}^{t}\left(u\right)=\hat{\Lambda }\left(t-u\right).$$

Then, we can write (Equation 29) in the form
(31)$$W\left(t\right)=\int_{-\infty }^{t}\Sigma \left(u\right)\cdot \dot{\hat{\Lambda }}\left(u\right)du$$
and treat the quantity $\hat{\Lambda }$ as the independent field variable. Let
(32)$${\hat{\Lambda }}_{r}^{t}\left(s\right)=\hat{\Lambda }\left(t\right)-{\hat{\Lambda }}^{t}\left(s\right)=\int_{t-s}^{t}\Lambda \left(u\right)du.$$

Note that the relative summed history ${\hat{\Lambda }}_{r}^{t}$ has the opposite sign to $\Lambda_{r}^{t}$ defined by (Equation 24), which is a choice that is more convenient in this context.

Dependence on $\hat{\Lambda }\left(t\right)$ cannot occur in the free energy or any other physical quantity. This is because the summed past history depends on the choice of the origin of the time variable. Thus,
(33)$$\psi \left(t\right)=\hat{\psi }\left({\hat{\Lambda }}_{r}^{t},\hat{\Lambda }\left(t\right)\right)=\hat{\psi }\left({\hat{\Lambda }}_{r}^{t}\right).$$

In addition, ϕ(t) drops out (see (Equation 28)) and
(34)$$\psi \left(t\right)={\hat{\psi }}_{r}\left({\hat{\Lambda }}_{r}^{t}\right).$$

Property 1 as given by (Equation 25) simplifies to
(35)$$D_{\hat{\Lambda }}{\hat{\psi }}_{r}\left({\hat{\Lambda }}_{r}^{t}\right)=\hat{\Sigma }\left({\hat{\Lambda }}_{r}^{t}\right)=\Sigma \left(t\right)$$
while (Equation 10) in Property 3 is replaced by
$$\dot{\psi }\left(t\right)\leq \dot{W}\left(t\right)=\Sigma \left(t\right)\cdot \Lambda \left(t\right).$$

Property 2 is given by (Equation 9) where the quantity ϕ is zero.

### 2.3. Minimal States

We now introduce the concept of a minimal state. This is an equivalence class of histories defined as follows [20,28,29,30,31]. The state of our system at a fixed time *t* is specified by the history and present value $S^{t}=\left(\Lambda^{t},\Lambda \left(t\right)\right)$. Let two states $S_{1}^{t}=\left(\Lambda_{1}^{t},\Lambda_{1}\left(t\right)\right)$, $S_{2}^{t}=\left(\Lambda_{2}^{t},\Lambda_{2}\left(t\right)\right)$ have the property that
(36)$$\tilde{\Sigma }\left(\Lambda_{1}^{t+s},\Lambda_{1}\left(t+s\right)\right)=\tilde{\Sigma }\left(\Lambda_{2}^{t+s},\Lambda_{2}\left(t+s\right)\right),\;\;\;\;s\in R^{+},$$
(37)$${\dot{\Lambda }}_{1}^{t+s}\left(u\right)={\dot{\Lambda }}_{2}^{t+s}\left(u\right),\;\;\;\;0\leq u\leq s,$$
then $S_{1}^{t}$, $S_{2}^{t}$ are said to be in the same equivalence class or minimal state. The latter terminology was introduced in [20]. Thus, if they have the same output from time *t* onwards, they are equivalent histories. The derivatives in (Equation 37) arise from the definition of a process in terms of the independent field variable ([14] for example). Requirement (Equation 37) means that $\Lambda_{1}\left(u\right)$, $\Lambda_{2}\left(u\right)$ differ by at most a constant for u≥t. Note that, for two equivalent states, we have
$$W_{1}\left(t,t+s\right)=W_{2}\left(t,t+s\right),\;\;\;\;s\in R$$
where $W_{1}$ is the quantity defined by (Equation 7) for $S_{1}^{t}$ and $W_{2}$ is that quantity for $S_{2}^{t}$.

**Remark** **2.**
*A fundamental distinction in the present work is whether the material under discussion has minimal states that are singletons, i.e., $S^{t}=\left(\Lambda^{t},\Lambda \left(t\right)\right)$ is the minimal state, or they contain more than one member.*


Observe that property (Equation 36) requires that
(38)$$\frac{\partial }{\partial \Lambda_{1}\left(t+s\right)}\psi \left(\Lambda_{1}^{t+s},\Lambda_{1}\left(t+s\right)\right)=\frac{\partial }{\partial \Lambda_{2}\left(t+s\right)}\psi \left(\Lambda_{2}^{t+s},\Lambda_{2}\left(t+s\right)\right),\;\;\;\;s\in R^{+}$$
for $S_{1}^{t}$, $S_{2}^{t}$ in the same minimal state.

## 3. Linear and Nonlinear Memory Models

We now consider free energies with quadratic memory terms, which produce linear memory constitutive equations. Let
(39)$$\begin{array}{cc} \psi (t) & =\phi \left(t\right)+\frac{1}{2}\int_{0}^{\infty }\int_{0}^{\infty }\Lambda_{r}^{t}\left(s_{1}\right)\cdot I\;K\left(s_{1},s_{2}\right)\Lambda_{r}^{t}\left(s_{2}\right)ds_{1}ds_{2}.\\ I\;K(s_{1},s_{2}) & ={I\;L}_{12}\left(s_{1},s_{2}\right)=\frac{\partial }{\partial s_{1}}\frac{\partial }{\partial s_{2}}I\;L\left(s_{1},s_{2}\right),\;\;\;\;I\;K,\;I\;L\in Lin\left(V\right). \end{array}$$
where IK,IL∈Lin(V). There is no loss of generality in taking
(40)$$I\;K\left(s_{1},s_{2}\right)={I\;K}^{⊤}\left(s_{2},s_{1}\right),\;\;\;\;\forall \;s_{1},s_{2}\in R^{+}.$$

Assuming that IK is integrable on $R^{+}\times R^{+}$, we have
(41)$$I\;L\left(s,u\right)=\int_{s}^{\infty }\int_{u}^{\infty }I\;K\left(s_{1},s_{2}\right)ds_{1}ds_{2},\;\;\;\;s,u\in R^{+}.$$

Applying Fubini’s theorem and (Equation 40), it follows that $I\;L\left(s,u\right)={I\;L}^{⊤}\left(u,s\right)$. In addition,
(42)$$\begin{array}{c} \lim_{s\to \infty }I\;L\left(s,u\right)=0,\;\;\;\;u\in R^{+},\;\;\;\;\;\;\;\;\lim_{s\to \infty }\frac{\partial }{\partial u}I\;L\left(s,u\right)=0,\;\;\;\;u\in R^{+} \end{array}$$
with similar limits holding at large *u* for fixed *s*. An alternative form of (Equation 39) is
(43)$$\psi \left(t\right)=\phi \left(t\right)+\frac{1}{2}\int_{0}^{\infty }\int_{0}^{\infty }{\dot{\Lambda }}^{t}\left(s_{1}\right)\cdot I\;L\left(s_{1},s_{2}\right){\dot{\Lambda }}^{t}\left(s_{2}\right)ds_{1}ds_{2},$$
(44)$${\dot{\Lambda }}^{t}\left(s\right)=\frac{d}{dt}\Lambda^{t}\left(s\right)=-\frac{d}{ds}\Lambda^{t}\left(s\right).$$

The form (Equation 39) emerges by expanding the general functional in (Equation 27) to include quadratic terms and neglecting any dependence on Λ(t) in the kernel ([23], for example and [1], p. 149). The linear term is omitted because it is inconsistent with the requirement that ψ be positive definite. The quantity ψ will be a valid free energy provided certain conditions are imposed on the kernel ${I\;L}_{12}$, which in particular must be a non-negative operator so that the second term on the right of (Equation 43) is non-negative.

Noting (Equation 41), we define
(45)$${I\;L}_{0}\left(u\right)=I\;L\left(0,u\right)={I\;L}^{⊤}\left(u,0\right),\;{I\;L}_{0}^{\prime }\left(u\right)=-\int_{0}^{\infty }I\;K\left(s_{1},u\right)ds_{1},$$
where the prime indicates differentiation with respect to the argument. The constitutive relation has the form
(46)$$\begin{array}{c} \Sigma \left(t\right)={\tilde{\Sigma }}_{e}\left(\Lambda \left(t\right)\right)+\int_{0}^{\infty }{I\;L}_{0}^{\prime }\left(u\right)\Lambda_{r}^{t}\left(u\right)du={\tilde{\Sigma }}_{e}\left(\Lambda \left(t\right)\right)+\int_{0}^{\infty }{I\;L}_{0}\left(u\right){\dot{\Lambda }}^{t}\left(u\right)du, \end{array}$$
where
(47)$${\tilde{\Sigma }}_{e}\left(\Lambda \left(t\right)\right)=\Sigma_{e}\left(t\right)=\frac{d\phi \left(\Lambda (t)\right)}{d\Lambda (t)}.$$

Causality requires that ${I\;L}_{0}$ vanishes on $R^{--}$ [39]. An alternative form of (Equation 46) is
(48)$$\tilde{\Sigma }\left(t\right)=\Sigma_{0}\left(t\right)+\int_{0}^{\infty }{I\;L}_{0}^{\prime }\left(u\right)\Lambda^{t}\left(u\right)du,\;\;\;\;\Sigma_{0}\left(t\right)=\Sigma_{e}\left(t\right)+{I\;L}_{0}\left(0\right)\Lambda \left(t\right).$$

The standard choice for ϕ(t) is given by
$$\phi \left(t\right)=\frac{1}{2}\Lambda \left(t\right)\cdot {I\;L}_{e}\Lambda \left(t\right),$$
so that from (Equation 47), we obtain $\Sigma_{e}\left(t\right)={I\;L}_{e}\Lambda \left(t\right)$. Thermodynamic arguments can be used to show that
$${I\;L}_{e}={I\;L}_{e}^{⊤},\;{I\;L}_{0}\left(0\right)={I\;L}_{0}^{⊤}\left(0\right),$$
using an adaption of a technique described in [14], for example. Here, the second property is a consequence of (Equation 45). In earlier work on free energies, involving tensor constitutive relations [15,18,19,20,22,23,40], it is also assumed that
(49)$${I\;L}_{0}\left(u\right)={I\;L}_{0}^{⊤}\left(u\right),\;\;\;\;u\in R^{+},$$
a condition which cannot be deduced from (Equation 40) or from thermodynamics.

Note that (Equation 39) can be put in the form
(50)$$\psi \left(t\right)=S\left(t\right)+\frac{1}{2}\int_{0}^{\infty }\int_{0}^{\infty }\Lambda^{t}\left(s\right)\cdot I\;K\left(s_{1},s_{2}\right)\Lambda^{t}\left(s_{2}\right)ds_{1}ds_{2},$$
(51)$$S\left(t\right)=\Lambda \left(t\right)\cdot \Sigma \left(t\right)-\frac{1}{2}\Lambda \left(t\right)\cdot {I\;L}_{0}\left(0\right)\Lambda \left(t\right).$$

It follows that
(52)$$\frac{\partial \psi (t)}{\partial \Lambda (t)}=\frac{\partial S(t)}{\partial \Lambda (t)}=\Sigma \left(t\right),\;\;\;\;t\in R.$$

By differentiating (Equation 43) with respect to *t* and using (Equation 11), we obtain [17,23]
(53)$$\begin{array}{cc} D(t)= & \frac{1}{2}\int_{0}^{\infty }\int_{0}^{\infty }{\dot{\Lambda }}^{t}\left(s_{1}\right)\cdot I\;D\left(s_{1},s_{2}\right){\dot{\Lambda }}^{t}\left(s_{2}\right)ds_{1}ds_{2}\\ I\;D(s_{1},s_{2})= & -\frac{\partial }{\partial s_{1}}I\;L\left(s_{1},s_{2}\right)-\frac{\partial }{\partial s_{2}}I\;L\left(s_{1},s_{2}\right). \end{array}$$

Thus, because of (Equation 10), ID must be a non-negative operator.

It is assumed that
(54)$${I\;L}_{0}^{\prime }\in C^{1}\cap L^{2}\cap L^{1}\left(R^{+};Lin\left(V\right)\right),$$
so that the Fourier transform of ${I\;L}^{\prime }$ exists. We have ([1], page 161)
(55)$${I\;L}_{0+}^{\prime }\left(\omega \right)=\int_{0}^{\infty }{I\;L}_{0}^{\prime }\left(s\right)e^{-i\omega s}ds={I\;L}_{0c}^{\prime }\left(\omega \right)-i\;{I\;L}_{0s}^{\prime }\left(\omega \right),$$
where ${I\;L}_{0c}^{\prime }$ and ${I\;L}_{0s}^{\prime }$ are the Fourier cosine and sine transforms. The latter vanishes at ω=0. It is a consequence of the second law that ([14,36], for example)
(56)$${I\;L}_{0s}^{\prime }\left(\omega \right)<0\;,\;\;\;\;\forall \;\omega \in R^{++},$$
for dissipative materials.

### Nonlinear Models

Let $\psi_{i}\left(t\right)$, i=0,1,2, be given as in (Equation 39),
$$\begin{array}{cc} \psi_{i}\left(t\right) & =\phi_{i}\left(t\right)+\psi_{i(r)}\left(t\right),\;\phi_{i}\left(t\right)={\tilde{\phi }}_{i}\left(\Lambda \left(t\right)\right)\\ \psi_{i(r)}\left(t\right) & =\frac{1}{2}\int_{0}^{\infty }\int_{0}^{\infty }\Lambda_{r}^{t}\left(s_{1}\right)\cdot {I\;K}^{\left(i\right)}\left(s_{1},s_{2}\right)\Lambda_{r}^{t}\left(s_{2}\right)ds_{1}ds_{2}, \end{array}$$
and satisfy Properties 1–3. The simplest nonlinear model is obtained from a quantity of the form
(57)$$\psi \left(t\right)=\psi_{0}\left(t\right)+\psi_{1}\left(t\right)\psi_{2}\left(t\right)$$
which is a free energy by Proposition 1. In particular, taking into account that
$$\Lambda_{r}^{t}\left(s_{1}\right)\cdot {I\;K}^{\left(i\right)}\left(s_{1},s_{2}\right)\Lambda_{r}^{t}\left(s_{2}\right)={I\;K}^{\left(i\right)}\left(s_{1},s_{2}\right):\left[\Lambda_{r}^{t}\left(s_{1}\right)⊗\Lambda_{r}^{t}\left(s_{2}\right)\right],$$
where ⊗ denotes the dyadic product, we can write
$$\begin{array}{cc} \psi (t) & =\phi_{0}\left(t\right)+\phi_{1}\left(t\right)\phi_{2}\left(t\right)+\psi_{0(r)}\left(t\right)+\phi_{1}\left(t\right)\psi_{2(r)}\left(t\right)+\phi_{2}\left(t\right)\psi_{1(r)}\left(t\right)\\ & +\frac{1}{4}\int_{R_{+}^{4}}{I\;K}^{\left(1\right)}\left(s_{1},s_{2}\right)⊗{I\;K}^{\left(2\right)}\left(s_{3},s_{4}\right):\overset{4}{\underset{j=1}{⊗}}\Lambda_{r}^{t}\left(s_{j}\right)d^{4}s, \end{array}$$
where $R_{+}^{n}={\left(R^{+}\right)}^{n}$, n∈IN,
$$d^{n}s=ds_{1}ds_{2}...ds_{n},\;\overset{n}{\underset{j=1}{⊗}}\Lambda_{r}^{t}\left(s_{j}\right)=\Lambda_{r}^{t}\left(s_{1}\right)⊗\Lambda_{r}^{t}\left(s_{2}\right)⊗...⊗\Lambda_{r}^{t}\left(s_{n}\right),$$
and ${I\;K}^{\left(1\right)}⊗{I\;K}^{\left(2\right)}$ is a fourth-order tensor belonging to Lin(Lin(V)). With a little abuse of notation, the double dot here denotes the scalar product in Lin(Lin(V)).

From (Equation 14), we have
(58)$$\Sigma \left(t\right)=\Sigma_{0}\left(t\right)+\psi_{2}\left(t\right)\Sigma_{1}\left(t\right)+\psi_{1}\left(t\right)\Sigma_{2}\left(t\right),$$
where $\Sigma_{i}$ denotes the dependent field related to $\psi_{i}$, i=0,1,2, which is given by
$$\begin{array}{cc} \Sigma_{i}\left(t\right) & ={\tilde{\Sigma }}_{e}^{\left(i\right)}\left(\Lambda \left(t\right)\right)+\int_{0}^{\infty }{\left[{I\;L}_{0}^{\left(i\right)}\right]}^{\prime }\left(u\right)\Lambda_{r}^{t}\left(u\right)du\\ & ={\tilde{\Sigma }}_{e}^{\left(i\right)}\left(\Lambda \left(t\right)\right)-\int_{0}^{\infty }\int_{0}^{\infty }{I\;K}^{\left(i\right)}\left(s_{1},u\right)\Lambda_{r}^{t}\left(u\right)\;ds_{1}du. \end{array}$$

The special case where $\psi_{1}=\psi_{2}$ and $\Sigma_{1}=\Sigma_{2}$ is the basis of developments in Section 5.1 and other sections.

Moreover, according to (Equation 17), the nonlinear rate of dissipation has the form
$$D\left(t\right)=D_{0}\left(t\right)+D_{1}\left(t\right)\psi_{2}\left(t\right)+D_{2}\left(t\right)\psi_{1}\left(t\right)$$
where
$$\begin{array}{cc} D_{i}\left(t\right)= & \frac{1}{2}\int_{0}^{\infty }\int_{0}^{\infty }{\dot{\Lambda }}^{t}\left(s_{1}\right)\cdot {I\;D}^{\left(i\right)}\left(s_{1},s_{2}\right){\dot{\Lambda }}^{t}\left(s_{2}\right)ds_{1}ds_{2}\\ {I\;D}^{\left(i\right)}\left(s_{1},s_{2}\right)= & -\frac{\partial }{\partial s_{1}}{I\;L}^{\left(i\right)}\left(s_{1},s_{2}\right)-\frac{\partial }{\partial s_{2}}{I\;L}^{\left(i\right)}\left(s_{1},s_{2}\right), \end{array}$$

${I\;L}^{\left(i\right)}$ being related to ${I\;K}^{\left(i\right)}$ as indicated by (Equation 41).

A more general expression, say $\psi_{nl}$, can be obtained by the functional Taylor expansion of $\tilde{\phi }$ and ${\hat{\psi }}_{r}$ in (Equation 27) and neglecting the third-order terms because $\psi_{nl}$ must be non-negative. We let
(59)$$\begin{array}{cc} \psi_{nl}\left(t\right) & ={\tilde{\phi }}_{nl}\left(\Lambda \left(t\right)\right)+\frac{1}{2}\int_{R_{+}^{2}}H_{0}\left(\Lambda \left(t\right),s_{1},s_{2}\right):\overset{2}{\underset{j=1}{⊗}}\Lambda_{r}^{t}\left(s_{j}\right)d^{2}s\\ \; & +\frac{1}{4}\int_{R_{+}^{4}}H\left(s_{1},s_{2},s_{3},s_{4}\right):\overset{4}{\underset{j=1}{⊗}}\Lambda_{r}^{t}\left(s_{j}\right)d^{4}s \end{array}$$
where $H_{0}$ and H are second and fourth-order tensors on V, respectively. Any dependence of H on Λ(t) is neglected.

Further constraints must be placed on $H_{0}$ and H to ensure that $\psi_{nl}$ has the required Properties 1–3 of a free energy. Here, we will limit ourselves to observing that (Equation 57) is recovered from (Equation 59) by letting
$$\phi_{nl}\left(t\right)=\phi_{0}\left(t\right)+\phi_{1}\left(t\right)\phi_{2}\left(t\right),\;H\left(s_{1},s_{2},s_{3},s_{4}\right)={I\;K}^{\left(1\right)}\left(s_{1},s_{2}\right)⊗{I\;K}^{\left(2\right)}\left(s_{3},s_{4}\right),$$
$$H_{0}\left(\Lambda \left(t\right),s_{1},s_{2}\right)={I\;K}^{\left(0\right)}\left(s_{1},s_{2}\right)+{\tilde{\phi }}_{1}\left(\Lambda \left(t\right)\right){I\;K}^{\left(2\right)}\left(s_{1},s_{2}\right)+{\tilde{\phi }}_{2}\left(\Lambda \left(t\right)\right){I\;K}^{\left(1\right)}\left(s_{1},s_{2}\right).$$

When summed past histories are involved, ψ reduces to $\psi_{r}$, as stated in (Equation 34), and therefore, we can simply choose
(60)$$\begin{array}{cc} \psi_{nl}\left(t\right) & =\frac{1}{2}\int_{R_{+}^{2}}H_{0}\left(s_{1},s_{2}\right):\overset{2}{\underset{j=1}{⊗}}{\hat{\Lambda }}_{r}^{t}\left(s_{j}\right)d^{2}s\\ \; & \;\;\;\;\;\;\;\;+\frac{1}{4}\int_{R_{+}^{4}}H\left(s_{1},s_{2},s_{3},s_{4}\right):\overset{4}{\underset{j=1}{⊗}}{\hat{\Lambda }}_{r}^{t}\left(s_{j}\right)d^{4}s, \end{array}$$
where ${\hat{\Lambda }}_{r}^{t}$ denotes the reduced summed history. In particular, (Equation 57) is recovered provided that
$$H_{0}={I\;K}^{\left(0\right)},\;H\left(s_{1},s_{2},s_{3},s_{4}\right)={I\;K}^{\left(1\right)}\left(s_{1},s_{2}\right)⊗{I\;K}^{\left(2\right)}\left(s_{3},s_{4}\right).$$

## 4. Minimal States and Quadratic Free Energies

Let us consider the concept of a minimal state in the context of linear memory constitutive equations. Applying the definition (Equation 36) to (Equation 46), we find [23] that $\left(\Lambda_{1}^{t},\Lambda_{1}\left(t\right)\right)$ and $\left(\Lambda_{2}^{t},\Lambda_{2}\left(t\right)\right)$ are equivalent, or in the same minimal state, if and only if
(61)$$\Lambda_{1}\left(t+s\right)=\Lambda_{2}\left(t+s\right),\;\;\;\;s\in R^{+}$$
provided the equilibrium quantity ${\tilde{\Sigma }}_{e}$ has a unique inverse, and
(62)$$I^{t}\left(s,\Lambda_{1}^{t}\right)=I^{t}\left(s,\Lambda_{2}^{t}\right),\;\;\;\;s\in R^{+}$$
where $I^{t}$ is the linear functional [16,20,25,30,31]
(63)$$I^{t}\left(s,\Lambda^{t}\right)=\int_{0}^{\infty }{I\;L}_{0}^{\prime }\left(s+u\right)\Lambda^{t}\left(u\right)du.$$

In the case where $\Sigma_{e}$ vanishes (see (Equation 33)), which is the case of primary interest here, there is no requirement that (Equation 61) holds, although
(64)$${\dot{\Lambda }}_{1}\left(t+s\right)={\dot{\Lambda }}_{2}\left(t+s\right),\;\;\;\;s\in R^{+}$$
must always be true. We introduce the relation
(65)$$\Lambda_{1}\left(t\right)=\Lambda_{2}\left(t\right)$$
as an extra condition in the definition of a minimal state. It follows from (Equation 64) that (Equation 61) holds. We shall sometimes refer to the equivalence or otherwise of histories, omitting the mention of present values, when the former are central to the argument.

The linearity of the functional $I^{t}$ means that the requirement of the equivalence of $\Lambda_{1}^{t}$ and $\Lambda_{2}^{t}$ is the same as that $\Lambda_{1}^{t}-\Lambda_{2}^{t}$ be equivalent to the zero history. Thus, if the minimal state including the zero history is singleton (non-singleton), then all minimal states are singleton (non-singleton).

In the arguments that follow, we introduce certain results obtained for the minimum, maximum and other free energies in [15,17,20] and related work, without developing the detailed apparatus.

Let $\psi_{r}$ in (Equation 27) have the form
(66)$$\psi_{r}\left(t\right)=\int_{-\infty }^{\infty }{\left|f(\omega ,\Lambda_{r}^{t})\right|}^{2}d\omega$$
where f:R→C is a linear functional of the history $\Lambda_{r}^{t}$ with the property that
(67)$$f\left(\omega ,\Lambda_{1r}^{t}\right)=f\left(\omega ,\Lambda_{2r}^{t}\right),\;\;\;\;\forall \omega \in R,$$
if and only if $\Lambda_{1r}^{t}$ and $\Lambda_{2r}^{t}$ are equivalent histories. The quantity f is thus a functional of the minimal state. We have
(68)$$f\left(\omega ,\Lambda^{t}\right)=0,\;\;\;\;\omega \in R\;⟺\;\;\;\;\;I^{t}\left(s,\Lambda^{t}\right)=0,\;\;\;\;s\in R^{+}.$$

If states are equivalent to the zero state, usually in the context of the difference of two equivalent states, the present value is zero, and a distinction between actual and relative histories is unnecessary.

**Remark** **3.***The form* (Equation 66) *applies to the minimum, maximum and a family of intermediate free energies given in [15,17,20,21]. In these cases, $f(\omega ,\Lambda_{r}^{t})$ is defined on the frequency domain.*

Let
(69)$$f\left(\omega ,\Lambda_{r}^{t}\right)=\int_{0}^{\infty }I\;U\left(\omega ,u\right)\Lambda_{r}^{t}\left(u\right)du,$$
so that
(70)$${I\;L}_{12}\left(u_{1},u_{2}\right)=\int_{-\infty }^{\infty }{I\;U}^{*}\left(\omega ,u_{1}\right)I\;U\left(\omega ,u_{2}\right)d\omega ,$$
where ${I\;U}^{*}$ is the Hermitian conjugate of IU. It follows from (Equation 66) and (Equation 67) that the free energy is itself a functional of the minimal state so that if $\left(\Lambda_{1r}^{t},\Lambda_{1}\left(t\right)\right)$, $\left(\Lambda_{2r}^{t},and\Lambda_{2}\left(t\right)\right)$ are equivalent states, then
(71)$$\begin{array}{cc} \hat{\psi }\left(\Lambda_{1r}^{t},\Lambda_{1}\left(t\right)\right) & =\hat{\psi }\left(\Lambda_{2r}^{t},\Lambda_{2}\left(t\right)\right)\\ \Lambda_{1}\left(t\right) & =\Lambda_{2}\left(t\right). \end{array}$$

Relation (Equation 38) follows automatically, but it is not necessary to assume (Equation 71) for this relation to be true. Let us introduce the scalar product notation
(72)$$\langle \Lambda_{1}^{t},\Lambda_{2}^{t}\rangle =\frac{1}{2}\int_{0}^{\infty }\int_{0}^{\infty }\Lambda_{1}^{t}\left(s_{1}\right)\cdot {I\;L}_{12}\left(s_{1},s_{2}\right)\Lambda_{2}^{t}\left(s_{2}\right)ds_{1}ds_{2}=\langle \Lambda_{2}^{t},\Lambda_{1}^{t}\rangle$$

The free energy is given by
(73)$$\psi \left(t\right)=\phi \left(t\right)+\langle \Lambda_{r}^{t},\Lambda_{r}^{t}\rangle =S\left(t\right)+\langle \Lambda^{t},\Lambda^{t}\rangle$$
where *S* is defined by (Equation 51).

We now prove certain results for free energies, using this bracket notation ([1], p. 173).

**Proposition** **2.**
*If the free energy is a functional of the minimal state and if $\Lambda_{1}^{t}$, $\Lambda_{2}^{t}$ are equivalent histories, then*

(74)
$$\langle \Lambda_{1}^{t},\Lambda_{1}^{t}\rangle =\langle \Lambda_{2}^{t},\Lambda_{2}^{t}\rangle =\langle \Lambda_{1}^{t},\Lambda_{2}^{t}\rangle ,\;\;\;\;t\in R^{+}.$$



**Proof.** The first equality in (Equation 74) follows from the definition of equivalence, on noting that *S*, and more obviously ϕ, are equal for the states $\left(\Lambda_{1}^{t},\Lambda \left(t\right)\right)$, $\left(\Lambda_{2}^{t},\Lambda \left(t\right)\right)$ at time $t,\;\;s\in R^{+}$. We also have
(75)$$\langle \Lambda_{d}^{t},\Lambda_{d}^{t}\rangle =0,\;\;\;\;\Lambda_{d}^{t}=\Lambda_{1}^{t}-\Lambda_{2}^{t}.$$
since $\Lambda_{d}^{t}$ is equivalent to the zero state. Thus, the last equality in (Equation 74) can be deduced using the bilinearity of the scalar product. □

It follows from Proposition 2 that
(76)$$\langle \Lambda_{1}^{t},\Lambda_{d}^{t}\rangle =\langle \Lambda_{2}^{t},\Lambda_{d}^{t}\rangle =0.$$

**Proposition** **3.***For a free energy with a history-dependent part of the form* (Equation 66), *the statement that $\langle \Lambda^{t},\Lambda^{t}\rangle$, is positive semi-definite; i.e., it vanishes for some non-zero $\Lambda_{1}^{t}$, and it is true if and only if the minimal states are non-singleton.*

**Proof.** If, for the non-zero history $\Lambda_{1}^{t}$, the quantity $I^{t}\left(u,\Lambda^{t}\right)$ vanishes for u≥0; in other words, if $\Lambda_{1}^{t}$ is equivalent to the zero history and the minimal states are non-singleton, then, from (Equation 75), $\langle \Lambda^{t},\Lambda^{t}\rangle$ vanishes at $\Lambda_{1}^{t}$ and is non-negative.If $\langle \Lambda^{t},\Lambda^{t}\rangle$ vanishes for the non-zero history $\Lambda_{1}^{t}$, then, from (Equation 66), $f(\omega ,\Lambda^{t})=0$ and by (Equation 68), we have that $I^{t}\left(s,\Lambda^{t}\right),\;s\geq 0$ vanishes, and the minimal states are non-singleton since $\Lambda_{1}^{t}$ is non-zero. □

In [21,23] (and also [1], p. 168), materials are characterized by the singularity types in the complex frequency plane of the Fourier transform of the derivative of the relaxation function. If this quantity has only isolated singularities (corresponding to a relaxation function consisting of sums of decaying exponentials, possibly multiplying polynomials and trigonometric functions) then minimal states are non-singleton. If the singularities characterizing a material include branch cuts, then the minimal states are singletons [24] (see also [36], p. 499). This is the case of main interest in the present work.

We adopt a different viewpoint on free energies and constitutive equations in this work. The standard thermodynamical point of view is to specify a free energy and deduce a constitutive relation from this. Alternatively, an applications-oriented approach, which is now adopted, involves deciding on a constitutive equation and searching for a free energy that yields this relation. This latter step may not be easy.

### 4.1. Quadratic Free Energies for Singleton Materials

We make the assumption in the following sections that the materials are such that their minimal states are singletons. This implies that the free energies, at least in the categories specified in Proposition 3, are positive definite functionals of the history.

#### 4.1.1. The Graffi Free Energy

Let (Equation 46)–(Equation 47) be the constitutive relations of the dependent field on a general vector space. A corresponding free energy is the Graffi functional, which is given by
(77)$$\psi^{G}\left(t\right)=\phi \left(t\right)-\frac{1}{2}\int_{0}^{\infty }\Lambda_{r}^{t}\left(s\right)\cdot {I\;L}_{0}^{\prime }\left(s\right)\Lambda_{r}^{t}\left(s\right)ds.$$

It satisfies Properties 1–3 of a free energy only if
(78)$${I\;L}_{0}^{\prime }\left(s\right)<0,\;\;\;\;\;\;\;\;{I\;L}_{0}^{\prime \prime }\left(s\right)\geq 0\;\;\;\;\forall s\in R,$$
so that these conditions are assumed to hold. The rate of dissipation is
(79)$$D^{G}\left(t\right)=\frac{1}{2}\int_{0}^{\infty }\Lambda_{r}^{t}\left(s\right)\cdot {I\;L}_{0}^{\prime \prime }\left(s\right)\Lambda_{r}^{t}\left(s\right)ds\geq 0.$$

Equation (Equation 77) can be written in the form (Equation 39) as indicated on page 238 of [1].

Let us assume that
(80)$${I\;L}_{0}^{\prime \prime }\left(s\right)+\lambda {I\;L}_{0}^{\prime }\left(s\right)\geq 0,\;\;\;\;s\in R^{+}.$$

It will be true, for instance, if ${I\;L}_{0}^{\prime }\left(u\right)$ consists of sums (or integrals) of decaying exponentials multiplying non-negative coefficients (or a non-negative function) with dominant term proportional to $e^{-\lambda u}$. It follows that
(81)$$D^{G}\left(t\right)\geq \lambda \left[\psi \left(t\right)-\phi \left(t\right)\right].$$

The Graffi free energy is not in general a functional of the minimal state [30]. It is, however, a positive definite functional of the history, by virtue of the first inequality in (Equation 78) and a positive definite function of the present value by virtue of the assumption after (Equation 5).

#### 4.1.2. The Work Function

Recalling the first equality in (Equation 46) we put
$$\Sigma_{h}\left(t\right)=\int_{0}^{\infty }{I\;L}_{0}^{\prime }\left(u\right)\Lambda_{r}^{t}\left(u\right)du=\int_{-\infty }^{t}{I\;L}_{0}^{\prime }\left(t-s\right)\left(\Lambda \left(s\right)-\Lambda \left(t\right)\right)ds.$$

Using (Equation 47), the total work performed on the material, given by (Equation 6), can be expressed in the form
$$\begin{array}{cc} W(t) & =\int_{-\infty }^{t}\Sigma_{e}\left(u\right)\cdot \dot{\Lambda }\left(u\right)du+\int_{-\infty }^{t}\Sigma_{h}\left(u\right)\cdot \dot{\Lambda }\left(u\right)du\\ & =\int_{-\infty }^{t}\frac{d\phi \left(\Lambda (u)\right)}{d\Lambda (u)}\cdot \dot{\Lambda }\left(u\right)du+\int_{-\infty }^{t}\int_{-\infty }^{u}{I\;L}_{0}^{\prime }\left(u-s\right)\left(\Lambda \left(s\right)-\Lambda \left(u\right)\right)ds\cdot \dot{\Lambda }\left(u\right)du\\ & =\phi \left(t\right)+\frac{1}{2}\int_{-\infty }^{t}\int_{-\infty }^{t}\left(\Lambda \left(s\right)-\Lambda \left(t\right)\right)\cdot {I\;K}_{0}\left(|s-u|\right)\left(\Lambda \left(u\right)-\Lambda \left(t\right)\right)duds, \end{array}$$
where ${I\;K}_{0}\left(|s-u|\right)=\frac{\partial }{\partial s}\frac{\partial }{\partial u}\;{I\;L}_{0}\left(|s-u|\right)$. Then, we conclude that
(82)$$W\left(t\right)=\phi \left(t\right)+\frac{1}{2}\int_{0}^{\infty }\int_{0}^{\infty }\Lambda_{r}^{t}\left(s\right)\cdot {I\;K}_{0}\left(|s-u|\right)\Lambda_{r}^{t}\left(u\right)duds.$$

This is a special example of (Equation 39) with $I\;L\left(s,u\right)={I\;L}_{0}\left(|s-u|\right)$. However, ${I\;L}_{12}$ has singular delta function behavior [40] and is therefore not bounded. We emphasize that the work function obeys the properties of a free energy with zero dissipation rate, D(t)=0, $t\in R^{+}$ [33,41,42].

We denote by $\psi^{M}$ the work function, given by (Equation 82), since it is the maximum free energy for singleton materials, but it is not in general a functional of the minimal state [30]. It is a positive definite function of Λ(t) and a positive definite functional of the history, which is clear from its representation in the frequency domain [15]. In particular, for singleton materials, it can be shown that
$$\psi^{G}\left(t\right)\leq \psi^{M}\left(t\right).$$

## 5. Viscoelastic Systems with Memory

In the sequel, the vector space V is Sym, the subspace of symmetric second-order tensors on $R^{3}$. In addition, memory kernels take values in Lin(V)=Lin(Sym), which represents the space of fourth-order tensors. Let E∈V and T∈V denote the strain and the stress, respectively. Using the notation (Equation 1), $E^{t}$ denotes the strain history and $E^{†}$ denotes the constant strain history of value E(t),
$$E^{†}\left(s\right)=E\left(t\right)\;,\;s\geq 0\;.$$

A material is viscoelastic if the stress tensor ***T*** not only depends on the current value of the strain but also on its history: $$T\left(t\right)=\tilde{T}\left(E\left(t\right),E^{t}\right).$$

The dependence of ***T*** and ***E*** on the space variable x is understood but not written.

The linear constitutive equation for a viscoelastic body is given by
(83)$$\tilde{T}\left(E\left(t\right),E^{t}\right)=G_{0}E\left(t\right)+\int_{0}^{\infty }G^{\prime }\left(s\right)E^{t}\left(s\right)ds,$$
where the memory kernel $G^{\prime }:R^{+}\to Lin\left(V\right)$ is a summable and continuous fourth-order tensor-valued function. It is of interest to compare this with the more general relation (Equation 46). Let
$$G\left(t\right)=G_{0}+\int_{0}^{t}G^{\prime }\left(s\right)ds,\;G_{\infty }=G_{0}+\int_{0}^{\infty }G^{\prime }\left(s\right)ds.$$

We can rewrite (Equation 83) as
(84)$$\hat{T}\left(E\left(t\right),E_{r}^{t}\right)=G_{\infty }E\left(t\right)+\int_{0}^{\infty }G^{\prime }\left(s\right)E_{r}^{t}\left(s\right)ds,$$
where $E_{r}^{t}\left(s\right)=E^{t}\left(s\right)-E\left(t\right)$. Since $E_{r}^{t}$ corresponds to $\Lambda_{r}^{t}$ in (Equation 24) if Λ=E, then G corresponds to ${I\;L}_{0}$ in (Equation 46) with $\Sigma_{e}\left(t\right)=G_{\infty }E\left(t\right)$. Accordingly, $G_{\infty }$ corresponds to ${I\;L}_{e}$. The consequence of the second law stated by (Equation 56) takes the form
(85)$$G_{s}^{\prime }\left(\omega \right)=\int_{0}^{\infty }G^{\prime }\left(s\right)\sin\omega s\;ds<0\;\forall \;\omega \in R^{++},$$
in the present context. Moreover, from thermodynamic arguments [14], it follows that
$$G_{\infty }=G_{\infty }^{⊤},\;G_{0}=G_{0}^{⊤},\;G_{0}-G_{\infty }>0.$$

According to Properties 1–3 and (Equation 4), a functional $\hat{\psi }\left(E\left(t\right),E_{r}^{t}\right)$ is said to be a free energy for the (possibly nonlinear) stress functional $\hat{T}\left(E\left(t\right),E_{r}^{t}\right)$ if it fulfills: (86)$$\begin{array}{cc} \; & \frac{\partial }{\partial E(t)}\hat{\psi }\left(E\left(t\right),E_{r}^{t}\right)-D_{E}\;\hat{\psi }\left(E\left(t\right),E_{r}^{t}\right)=\hat{T}\left(E\left(t\right),E_{r}^{t}\right)\\ \; & \hat{\psi }\left(E\left(t\right),E_{r}^{t}\right)\geq \tilde{\phi }\left(E\left(t\right)\right)\;\mathrm{and}\;\tilde{\phi }\left(0\right)=0\\ \; & \dot{\psi }\left(t\right)\leq \dot{W}\left(t\right)=T\left(t\right)\cdot \dot{E}\left(t\right),\;\psi \left(t\right)=\hat{\psi }\left(E\left(t\right),E_{r}^{t}\right),\;T\left(t\right)=\hat{T}\left(E\left(t\right),E_{r}^{t}\right) \end{array}$$
for all $E\left(t\right),E_{r}^{t}$. The term $D_{E}\;\hat{\psi }$ is related to the Fréchét differential $\delta \hat{\psi }$ through the representation formula (see (Equation 26))
$$D_{E}\;\hat{\psi }\left(E\left(t\right),E_{r}^{t}\right)\cdot E\left(t\right)=\delta \hat{\psi }\left(E\left(t\right),E_{r}^{t}|E^{†}\right).$$

The free energy (Equation 39) becomes
(87)$$\psi \left(t\right):=\hat{\psi }\left(E\left(t\right),E_{r}^{t}\right)=\tilde{\phi }\left(E(t)\right)+\frac{1}{2}\int_{0}^{\infty }\int_{0}^{\infty }E_{r}^{t}\left(s_{1}\right)\cdot I\;K\left(s_{1},s_{2}\right)E_{r}^{t}\left(s_{2}\right)ds_{1}ds_{2},$$
where $I\;K\left(s_{1},s_{2}\right)={I\;L}_{12}\left(s_{1},s_{2}\right)$. Moreover, (Equation 11) takes the form
(88)$$\dot{\psi }\left(t\right)+D\left(t\right)=T\left(t\right)\cdot \dot{E}\left(t\right)$$
where D(t)≥0 is the rate of dissipation, which is given in general by (Equation 53) and here by
$$\begin{array}{cc} \; & D\left(t\right)=\frac{1}{2}\int_{0}^{\infty }\int_{0}^{\infty }{\dot{E}}^{t}\left(s_{1}\right)\cdot I\;D\left(s_{1},s_{2}\right){\dot{E}}^{t}\left(s_{2}\right)\;ds_{1}\;ds_{2},\\ \; & I\;D\left(s_{1},s_{2}\right)=-\frac{\partial }{\partial s_{1}}I\;L\left(s_{1},s_{2}\right)-\frac{\partial }{\partial s_{2}}I\;L\left(s_{1},s_{2}\right). \end{array}$$

Since Bearing in mind that
$${\dot{E}}^{t}\left(s\right)=-\frac{d}{ds}E^{t}\left(s\right)=-\frac{d}{ds}E_{r}^{t}\left(s\right)$$
a double integration by parts, with respect to $s_{1}$ and $s_{2}$, yields
$$\begin{array}{cc} & D\left(t\right)=\frac{1}{2}\int_{0}^{\infty }\int_{0}^{\infty }E_{r}^{t}\left(s_{1}\right)\cdot {I\;D}_{12}\left(s_{1},s_{2}\right)E_{r}^{t}\left(s_{2}\right)\;ds_{1}\;ds_{2},\\ & {I\;D}_{12}\left(s_{1},s_{2}\right)=\frac{\partial }{\partial s_{1}}\frac{\partial }{\partial s_{2}}I\;D\left(s_{1},s_{2}\right). \end{array}$$

We have the correspondences between G and ${I\;L}_{0}$ and between $G_{\infty }$ and ${I\;L}_{e}$ noted after (Equation 84). In addition, the work function (Equation 82) becomes
$${\hat{\psi }}^{M}\left(E\left(t\right),E_{r}^{t}\right)=\frac{1}{2}E\left(t\right)\cdot G_{\infty }E\left(t\right)+\frac{1}{2}\int_{0}^{\infty }\;\;\;\int_{0}^{\infty }E_{r}^{t}\left(s_{1}\right)\cdot G_{12}\left(\right|s_{1}-s_{2}\left|\right)E_{r}^{t}\left(s_{1}\right)ds_{1}ds_{2}.$$

Since ${\dot{\psi }}^{M}\left(t\right)=T\left(t\right)\cdot \dot{E}\left(t\right)$, it follows from (Equation 88) that $D^{M}\left(t\right)=0$.

Graffi’s free energy takes the form
(89)$${\hat{\psi }}^{G}\left(E\left(t\right),E_{r}^{t}\right)=\frac{1}{2}E\left(t\right)\cdot G_{\infty }E\left(t\right)-\frac{1}{2}\int_{0}^{\infty }E_{r}^{t}\left(s\right)\cdot G^{\prime }\left(s\right)E_{r}^{t}\left(s\right)ds.$$

In addition,
$$D^{G}\left(t\right)=\frac{1}{2}\int_{0}^{\infty }E_{r}^{t}\left(s\right)\cdot G^{\prime \prime }\left(s\right)E_{r}^{t}\left(s\right)\;ds.$$

For the functional specified by (Equation 89) to be a free energy, it is required that (see (Equation 78))
(90)$$G^{\prime }\left(s\right)<0,\;\;\;\;\;\;\;\;G^{\prime \prime }\left(s\right)\geq 0.$$

### 5.1. Nonlinear Constitutive Equations

Special cases of Proposition 1 are now considered in the context of viscoelasticity.

Let ${\hat{T}}_{\ell }$ be the given linear constitutive functional (Equation 84) and ${\hat{\psi }}_{\ell }$ any related free energy functional with kernel G. Let $f:R^{+}\to R^{+}$ be any given smooth function such that
(91)$$f\left(0\right)=0,\;f^{\prime }>0,\;f^{\prime }\left(0\right)=1.$$

Then, nonlinear stress–strain constitutive equations can be obtained by considering the memory relation
(92)$$T\left(t\right)=f^{\prime }\left(\psi_{\ell }\left(t\right)\right){\hat{T}}_{\ell }\left(E\left(t\right),E_{r}^{t}\right)$$
and the corresponding nonlinear free energy is
(93)$$\psi \left(t\right)=f\left({\hat{\psi }}_{\ell }\left(E\left(t\right),E_{r}^{t}\right)\right).$$

Indeed, we have ψ(t)≥0 and
$$T\left(t\right)\cdot \dot{E}\left(t\right)=f^{\prime }\left(\psi_{\ell }\left(t\right)\right)T_{\ell }\left(t\right)\cdot \dot{E}\left(t\right)\geq f^{\prime }\left(\psi_{\ell }\left(t\right)\right){\dot{\psi }}_{\ell }\left(t\right)=\dot{\psi }\left(t\right).$$

For example, we can choose $f\left(x\right)=x+x^{2}$, $f\left(x\right)=e^{x}-1$ and f(x)=log(x+1). When $f\left(x\right)=x+x^{2}$, then (Equation 92) and (Equation 93) yield
(94)$$\begin{array}{cc} T(t) & =\left(1+2\psi_{\ell }\left(t\right)\right)T_{\ell }\left(t\right),\\ \psi (t) & =\psi_{\ell }\left(t\right)\left(1+\psi_{\ell }\left(t\right)\right),\\ D(t) & =D_{\ell }\left(t\right)\left(1+2\psi_{\ell }\left(t\right)\right). \end{array}$$

In fact, we should replace 1 in the relation for ψ(t) by a constant with the dimensions of free energy so as to maintain explicitly correct dimensionality in each expression. What we are doing here is choosing units such that this constant has a value of 1.

More generally, let us consider memory kernels $G_{i},\;i=1,2,\ldots n$, satisfying (Equation 85) and $\lim_{s\to \infty }G_{i}\left(s\right)=G_{i\infty }$. Let
$$T_{i}\left(t\right)=G_{i\infty }E\left(t\right)+\int_{0}^{\infty }G_{i}^{\prime }\left(s\right)E_{r}^{t}\left(s\right)ds,$$
denote the related *i*-th linear model and let $\psi_{i}\left(t\right)$ be any associated free energy satisfying (Equation 86). Thus, we have
$$\psi_{i}\left(t\right)\geq 0\;\mathrm{and}\;{\dot{\psi }}_{i}\left(t\right)\leq T_{i}\left(t\right)\cdot \dot{E}\left(t\right).$$

In particular, for any given pair of kernels $G_{i}$ and $G_{j}$, we can construct a nonlinear stress–strain functional of the form
(95)$$T\left(t\right)=T_{i}\left(t\right)+2\psi_{j}\left(t\right)T_{j}\left(t\right).$$
which admits the following free energy functional
(96)$$\psi \left(t\right)=\psi_{i}\left(t\right)+{\left(\psi_{j}\left(t\right)\right)}^{2}.$$

In addition, given a nonlinear function *g* obeying the relations specified in (Equation 91), we can generalize (Equation 95) and (Equation 96) as follows: $$\begin{array}{c} T\left(t\right)=T_{i}\left(t\right)+g^{\prime }\left(\psi_{j}\left(t\right)\right)T_{j}\left(t\right),\;\psi \left(t\right)=\psi_{i}\left(t\right)+g\left(\psi_{j}\left(t\right)\right). \end{array}$$
for any fixed pair of integers i,j. By virtue of (Equation 88), we have
(97)$$\dot{\psi }\left(t\right)=T\left(t\right)\cdot \dot{E}\left(t\right)-D\left(t\right),\;D\left(t\right)=D_{i}\left(t\right)+g^{\prime }\left(\psi_{j}\left(t\right)\right)D_{j}\left(t\right)\geq 0.$$

### 5.2. A Nonlinear Viscoelastic Model Based on Graffi’s Free Energy

Letting $\psi_{\ell }={\hat{\psi }}^{G}$ and $T_{\ell }=\hat{T}$, as given by (Equation 89) and (Equation 84), respectively, we obtain from (Equation 94) a nonlinear constitutive equation of the following form
$$\begin{array}{cc} T(t)\; & =G_{\infty }E\left(t\right)+\int_{0}^{\infty }G^{\prime }\left(s\right)E_{r}^{t}\left(s\right)ds+\left[G_{\infty }E\left(t\right)⊗G_{\infty }E\left(t\right)\right]E\left(t\right)\\ \; & +E\left(t\right)\cdot G_{\infty }E\left(t\right)\int_{0}^{\infty }G^{\prime }\left(s\right)E_{r}^{t}\left(s\right)ds-\int_{0}^{\infty }E_{r}^{t}\left(s\right)\cdot G^{\prime }\left(s\right)E_{r}^{t}\left(s\right)ds\;G_{\infty }E\left(t\right)\\ \; & -\int_{0}^{\infty }\int_{0}^{\infty }\left[G^{\prime }\left(u\right)E_{r}^{t}\left(u\right)⊗G^{\prime }\left(s\right)E_{r}^{t}\left(s\right)\right]E_{r}^{t}\left(s\right)dsdu \end{array}$$
and a related free energy given by
$$\begin{array}{cc} \psi (t)\; & =\frac{1}{2}E\left(t\right)\cdot G_{\infty }E\left(t\right)-\frac{1}{2}\int_{0}^{\infty }E_{r}^{t}\left(s\right)\cdot G^{\prime }\left(s\right)E_{r}^{t}\left(s\right)ds\\ \; & +\frac{1}{4}{\left(E\left(t\right)\cdot G_{\infty }E\left(t\right)-\int_{0}^{\infty }E_{r}^{t}\left(s\right)\cdot G^{\prime }\left(s\right)E_{r}^{t}\left(s\right)ds\right)}^{2}. \end{array}$$

Moreover, the corresponding rate of dissipation is
$$D\left(t\right)=\frac{1}{2}\int_{0}^{\infty }E_{r}^{t}\left(s\right)\cdot G^{\prime \prime }\left(s\right)E_{r}^{t}\left(s\right)\;ds\left[1+E\left(t\right)\cdot G_{\infty }E\left(t\right)-\int_{0}^{\infty }E_{r}^{t}\left(s\right)\cdot G^{\prime }\left(s\right)E_{r}^{t}\left(s\right)ds\right],$$

For isotropic viscoelastic materials, the kernel $G^{\prime }$ and the relaxation modulus $G_{\infty }$ take the special form
$$G^{\prime }\left(s\right)=\lambda^{\prime }\left(s\right)1⊗1+2\mu^{\prime }\left(s\right)I,\;G_{\infty }=\lambda_{\infty }1⊗1+2\mu_{\infty }I,$$
where 1 is the unit second-order tensor and I the unit fourth-order tensor. Here $\lambda^{\prime },\mu^{\prime }\in C^{1}\cap L^{1}\left(R^{+};R\right)$, so that
$$\lambda \left(s\right)=\lambda_{0}+\int_{0}^{t}\lambda^{\prime }\left(s\right)\;ds,\;\mu \left(s\right)=\mu_{0}+\int_{0}^{t}\mu^{\prime }\left(s\right)\;ds,$$
and $\lambda_{\infty }=\lim_{s\to \infty }\lambda \left(s\right)$, $\mu_{\infty }=\lim_{s\to \infty }\mu \left(s\right)$. Moreover, $\lambda^{\prime },\mu^{\prime }<0$ and $\lambda^{\prime \prime },\mu^{\prime \prime }\geq 0$, which are the conditions (Equation 90). Using the decomposition,
$$E=\frac{1}{3}\left(\mathrm{tr}E\right)1+E_{\circ },\;T=\frac{1}{3}\left(\mathrm{tr}T\right)1+T_{\circ }$$
where tr stands for the trace and the subscript _∘_ denotes the deviatoric part of the tensor, we rewrite the nonlinear stress–strain relation in the form
$$\begin{array}{cc} T_{\circ }\left(t\right)\; & =2\mu_{\infty }E_{\circ }\left(t\right)[1+2\mu_{\infty }|E_{\circ }{\left(t\right)|}^{2}]+2[1+2\mu_{\infty }{\left|E_{\circ }\left(t\right)\right|}^{2}\int_{0}^{\infty }\mu^{\prime }\left(s\right)E_{\circ r}^{t}\left(s\right)ds\\ \; & -4\mu_{\infty }E_{\circ }\left(t\right)\int_{0}^{\infty }\mu^{\prime }\left(s\right)|E_{\circ r}^{t}{\left(s\right)|}^{2}ds-4\int_{0}^{\infty }\mu^{\prime }\left(u\right)E_{\circ r}^{t}\left(u\right)du\int_{0}^{\infty }\mu^{\prime }\left(s\right){\left|E_{\circ r}^{t}\left(s\right)\right|}^{2}ds, \end{array}$$
$$\begin{array}{cc} \; & \mathrm{tr}\;T\left(t\right)=K_{\infty }\mathrm{tr}\;E\left(t\right)[1+K_{\infty }{\left|\mathrm{tr}\;E\left(t\right)\right|}^{2}]+[1+K_{\infty }{\left|\mathrm{tr}\;E\left(t\right)\right|}^{2}\;\int_{0}^{\infty }\;K^{\prime }\left(s\right)\mathrm{tr}\;E_{r}^{t}\left(s\right)ds\\ \; & -K_{\infty }\mathrm{tr}\;E\left(t\right)\int_{0}^{\infty }K^{\prime }\left(s\right)|\mathrm{tr}\;E_{r}^{t}{\left(s\right)|}^{2}ds-\int_{0}^{\infty }K^{\prime }\left(u\right)\mathrm{tr}\;E_{r}^{t}\left(u\right)du\int_{0}^{\infty }K^{\prime }\left(s\right){\left|\mathrm{tr}\;E_{r}^{t}\left(s\right)\right|}^{2}ds, \end{array}$$
where $K\left(s\right)=\lambda \left(s\right)+\frac{2}{3}\mu \left(s\right)$ and $K_{\infty }=\lambda_{\infty }+\frac{2}{3}\mu_{\infty }$ denote the bulk elastic kernel and the bulk relaxation modulus, respectively.

The related free energy takes the form
$$\begin{array}{cc} \; & \psi \left(t\right)=\frac{1}{2}K_{\infty }{\left|\mathrm{tr}\;E\left(t\right)\right|}^{2}+\mu_{\infty }|E_{\circ }{\left(t\right)|}^{2}-\frac{1}{2}\int_{0}^{\infty }K^{\prime }\left(s\right)|\mathrm{tr}\;E_{r}^{t}{\left(s\right)|}^{2}ds-\int_{0}^{\infty }\mu^{\prime }\left(s\right){\left|E_{\circ r}^{t}\left(s\right)\right|}^{2}ds\\ \; & +{\left(\frac{1}{2}K_{\infty }{\left|\mathrm{tr}\;E\left(t\right)\right|}^{2}+\mu_{\infty }|E_{\circ }{\left(t\right)|}^{2}-\frac{1}{2}\int_{0}^{\infty }K^{\prime }\left(s\right)|\mathrm{tr}\;E_{r}^{t}{\left(s\right)|}^{2}ds-\int_{0}^{\infty }\mu^{\prime }\left(s\right){\left|E_{\circ r}^{t}\left(s\right)\right|}^{2}ds\right)}^{2}. \end{array}$$

### 5.3. A One-Dimensional Example

Consider one-dimensional models associated with strain and applied traction in the direction e so that
E=Ee⊗e,T=Te⊗e.

The symbol *T* for the component of ***T*** is consistent with the engineering stress considered in the literature as the ratio of the axial force over the reference area. Moreover, for simplicity, let
$$G^{\prime }\left(s\right)=-\alpha \left(G_{0}-G_{\infty }\right)\exp\left[-\alpha s\right]I,\;G_{0}=G_{0}I,\;G_{\infty }=G_{\infty }I,$$
where $\alpha ,G_{0},G_{\infty }>0$ and $G_{0}>G_{\infty }$. Letting $T_{\ell }=\hat{T}$, after differentiating equation (Equation 84) with respect to time, we obtain
(98)$${\dot{T}}_{\ell }\left(t\right)=G_{0}\dot{E}\left(t\right)-\alpha \left[T_{\ell }\left(t\right)-G_{\infty }E\left(t\right)\right],$$
which represents the well-known equation for a standard linear solid (or Zener model). The corresponding Graffi’s free energy $\psi_{\ell }={\hat{\psi }}^{G}$ satisfies a similar differential equation,
(99)$${\dot{\psi }}_{\ell }\left(t\right)=T_{\ell }\left(t\right)\dot{E}\left(t\right)-\alpha \left[\psi_{\ell }\left(t\right)-\frac{1}{2}G_{\infty }E^{2}\left(t\right)\right].$$

The parameter α represents the reciprocal of the characteristic relaxation time of the material. Equations (Equation 98) and (Equation 99) are not invariant under the time transformation
t→ct,c>0,
and hence, they describe a rate-dependent material behavior. In particular, they predict different linear elastic behavior as c→0 (very fast processes) and c→+∞ (very slow processes); since α→α/c we have (up to additive constants)
$$T_{\ell }\left(t\right)\to G_{\infty }E\left(t\right),\;\psi_{\ell }\left(t\right)\to \frac{1}{2}G_{\infty }E^{2}\left(t\right)\;\mathrm{as}\;c\to 0,$$
$$T_{\ell }\left(t\right)\to G_{0}E\left(t\right),\;\psi_{\ell }\left(t\right)\to \frac{1}{2}G_{0}E^{2}\left(t\right)\;\mathrm{as}\;c\to +\infty .$$

Accordingly, from (Equation 94), it follows
$$T\left(t\right)\to \left[1+G_{\infty }E^{2}\left(t\right)\right]G_{\infty }E\left(t\right)\;\mathrm{as}\;c\to 0$$
$$T\left(t\right)\to \left[1+G_{0}E^{2}\left(t\right)\right]G_{0}E\left(t\right),\;\mathrm{as}\;c\to +\infty ,$$

The asymptotic traction responses for *T* and $T_{\ell }$ is plotted in Figure 1 for both limit cases.

## 6. Electric Conductors with Memory

In this section, the vector space V is $R^{3}$. In addition, kernels take values in $Lin\left(V\right)=Lin\left(R^{3}\right)$, which is the space of second-order tensors. Let Sym(V) denote the subspace of Lin(V) which contains all symmetric tensors and $Sym^{+}\left(V\right)$ contains the convex set of positive-definite symmetric tensors (a tensor σ∈Sym(V) is positive definite if v·σv>0 for all non-zero v∈V).

In accordance with the notation specified by (Equation 1), let E(t) denote the electric field at time *t* and $E^{t}$ denote its past history. The constant history equal to E(t) is given by
$$E^{†}\left(s\right)=E\left(t\right)\;,\;s\geq 0\;.$$

An electric conductor is hereditary if the current vector J depends on the electric field history: $$J\left(t\right)=\tilde{J}\left(E^{t}\right).$$

As previously, the dependence of J and E on the space variable x is understood and not written.

We start from a basic, linear constitutive equation for the current
(100)$$\tilde{J}\left(E^{t}\right)=\int_{0}^{\infty }\sigma \left(s\right)E^{t}\left(s\right)ds$$
where the memory kernel $\sigma :R^{+}\to Sym^{+}\left(R^{3}\right)$ is a summable, continuous and positive-definite tensor-valued function. Let $\sigma_{0}=\int_{0}^{\infty }\sigma \left(s\right)ds\in Sym^{+}$. At any constant history $E^{†}$, we have
$$\tilde{J}\left(E^{†}\right)=\sigma_{0}\;E\left(t\right).$$

This relation resembles Ohm’s law and $\sigma_{0}$ is referred to as the relaxation conductivity tensor.

Note that the common form of Ohm’s law, $J\left(t\right)=\sigma_{0}\;E\left(t\right)$, is actually recovered if in (Equation 100) we (formally) choose the kernel σ equal to $\sigma_{0}$ times the Dirac mass at zero, $\delta_{0}$.

If we introduce the magnetic vector potential A and assume the vanishing of the electric scalar potential (as usual in electric conductors), we obtain
$$E\left(t\right)=-\dot{A}\left(t\right),\;A\left(s\right)-A\left(t\right)=\int_{s}^{t}E\left(u\right)\;du,$$
so that the relative past history of A, given by (Equation 24), equals the *relative summed past history* of the electric field in the notation (Equation 32), since
(101)$$A_{r}^{t}\left(s\right)=A\left(t-s\right)-A\left(t\right)=\int_{t-s}^{t}E\left(u\right)\;du=\int_{0}^{s}E^{t}\left(\tau \right)\;d\tau .$$

Thus,
(102)$$\frac{\partial }{\partial s}A_{r}^{t}\left(s\right)=E^{t}\left(s\right),\;\frac{\partial }{\partial t}A_{r}^{t}\left(s\right)=\dot{A}\left(t-s\right)-\dot{A}\left(t\right)=E\left(t\right)-E\left(t-s\right)=-E_{r}^{t}\left(s\right)$$
and after an integration by parts, we can rewrite (Equation 100) in the form
(103)$$\hat{J}\left(A_{r}^{t}\right)=\int_{0}^{\infty }\kappa \left(s\right)A_{r}^{t}\left(s\right)ds,$$
where $\kappa =-\sigma^{\prime }$. The history $A_{r}^{t}$, given by (Equation 101), corresponds to $\Lambda_{r}^{t}$ in (Equation 24) if Λ=A. In addition, we see that κ corresponds to ${I\;L}_{0}^{\prime }$ in (Equation 46) with $\Sigma_{e}=0$. Alternatively, taking Λ=E and $\hat{\Lambda }=-A$, then $A_{r}^{t}$ corresponds to ${\hat{\Lambda }}_{r}^{t}$; given by (Equation 32) and from (Equation 31), it follows that
$$W\left(t\right)=\int_{-\infty }^{t}\tilde{J}\left(E^{\tau }\right)\cdot E\left(\tau \right)d\tau =-\int_{-\infty }^{t}\hat{J}\left(A_{r}^{\tau }\right)\cdot \dot{A}\left(\tau \right)d\tau .$$

Condition (Equation 56), which is a consequence of the second law, takes the form [43]
(104)$$\kappa_{s}\left(\omega \right)=\int_{0}^{\infty }\kappa \left(s\right)\sin\omega s\;ds>0\;\forall \;\omega \in R^{++},$$
in the present context. It is satisfied if κ is positive-definite.

A functional $\hat{\psi }$ of the relative past history of the magnetic vector potential, $A_{r}^{t}$, is said to be a free energy for the (possibly nonlinear) current functional $\hat{J}\left(A_{r}^{t}\right)$, if it fulfills Properties 1–3 and (Equation 4), as adapted as in Section 2.1 and Section 2.2. We write these as follows: (105)$$\begin{array}{cc} \; & D_{A}\;\hat{\psi }\left(A_{r}^{t}\right)=-\hat{J}\left(A_{r}^{t}\right)\\ \; & \hat{\psi }\left(A_{r}^{t}\right)\geq 0\;\forall \;A_{r}^{t}\;\mathrm{and}\;\hat{\psi }\left(0^{†}\right)=0\\ \; & \dot{\psi }\left(t\right)\leq J\left(t\right)\cdot E\left(t\right)=-J\left(t\right)\cdot \dot{A}\left(t\right), \end{array}$$
where $\psi \left(t\right)=\hat{\psi }\left(A_{r}^{t}\right)$, $J\left(t\right)=\hat{J}\left(A_{r}^{t}\right)$. Note that we omit any dependence of $\hat{\psi }$ on A(t) as in (Equation 35). The term $D_{A}\;\hat{\psi }\left(A_{r}^{t}\right)$ is related to the Fréchét differential $\delta \hat{\psi }\left(A_{r}^{t}|\;\cdot \right)$ through the representation formula (see (Equation 26))
$$D_{A}\;\hat{\psi }\left(A_{r}^{t}\right)\cdot A\left(t\right)=\delta \tilde{\psi }\left(A_{r}^{t}|\;A^{†}\right)$$
for any choice of A(t). In the context of quadratic free energies, the operation $D_{A}$ is simply differentiation with respect to the explicit occurrence of A(t) in $A_{r}^{t}$ as given in (Equation 101).

Firstly, the general form of the free energy (Equation 39) reduces to
(106)$$\begin{array}{cc} \psi (t) & =\hat{\psi }\left(A_{r}^{t}\right)=\frac{1}{2}\int_{0}^{\infty }\int_{0}^{\infty }A_{r}^{t}\left(s_{1}\right)\cdot I\;K\left(s_{1},s_{2}\right)A_{r}^{t}\left(s_{2}\right)\;ds_{1}\;ds_{2}\\ \; & =\frac{1}{2}\int_{0}^{\infty }\int_{0}^{\infty }E^{t}\left(s_{1}\right)\cdot I\;L\left(s_{1},s_{2}\right)E^{t}\left(s_{2}\right)\;ds_{1}\;ds_{2}, \end{array}$$
where $I\;K\left(s_{1},s_{2}\right)={I\;L}_{12}\left(s_{1},s_{2}\right)$. Moreover, we can write (Equation 105)_4_ in the form: $$\dot{\psi }\left(t\right)+D\left(t\right)=J\left(t\right)\cdot E\left(t\right),$$
where D(t)≥0 is the rate of dissipation, which is given in general by (Equation 53) and here by
$$\begin{array}{cc} D(t)= & \frac{1}{2}\int_{0}^{\infty }\int_{0}^{\infty }E^{t}\left(s_{1}\right)\cdot I\;D\left(s_{1},s_{2}\right)E^{t}\left(s_{2}\right)\;ds_{1}\;ds_{2},\\ I\;D(s_{1},s_{2})= & -\frac{\partial }{\partial s_{1}}I\;L\left(s_{1},s_{2}\right)-\frac{\partial }{\partial s_{2}}I\;L\left(s_{1},s_{2}\right). \end{array}$$

Letting ${I\;K}_{0}=\kappa$ subject to (Equation 104), the work function (Equation 82) becomes
$$\begin{array}{cc} \psi^{M}\left(t\right) & ={\hat{\psi }}^{M}\left(A_{r}^{t}\right)=\frac{1}{2}\int_{0}^{\infty }\int_{0}^{\infty }A_{r}^{t}\left(s_{1}\right)\cdot \kappa (|s_{1}-s_{2}\left|\right)A_{r}^{t}\left(s_{2}\right)\;ds_{1}\;ds_{2}. \end{array}$$

There are many choices of free energy that can be used in this context. For example, there is the explicit form for the minimum free energy relating to continuous spectrum materials, which is derived in [24]. Minimal states are singletons for such materials. However, we will opt for algebraic simplicity by choosing the Graffi free energy (Equation 77). This takes the form
$$\psi^{G}\left(t\right)={\tilde{\psi }}^{G}\left(A_{r}^{t}\right)=\frac{1}{2}\int_{0}^{\infty }A_{r}^{t}\left(s\right)\cdot \kappa \left(s\right)A_{r}^{t}\left(s\right)ds.$$

For this functional to be a free energy, it is required that
$$\kappa \left(s\right)>0,\;\;\;\;\;\;\;\;\kappa^{\prime }\left(s\right)\leq 0.$$

### 6.1. Nonlinear Electric Conductors

We now consider special cases of Proposition 1 in the context of electrical conductors with memory.

Let ${\hat{J}}_{\kappa }$ be a given linear constitutive functional (Equation 103) and ${\tilde{\psi }}_{\kappa }$ any related Graffi free energy functional with kernel κ. Various forms of nonlinear constitutive equation can be obtained by taking the nonlinear current to be
(107)$$J\left(t\right)=f^{\prime }\left(\psi_{\kappa }\left(t\right)\right)J_{\kappa }\left(t\right),$$
where *f* satisfies (Equation 91). Due to the condition $f^{\prime }\left(0\right)=1$, the linear constitutive equation for $J_{\kappa }$ gives the first-order approximation to (Equation 107). The expression for the corresponding nonlinear free energy is
$$\psi \left(t\right)=f\left(\psi_{\kappa }\left(t\right)\right).$$

Indeed, we have ψ(t)≥0 and
$$J\left(t\right)\cdot E\left(t\right)=f^{\prime }\left(\psi_{\kappa }\left(t\right)\right)J_{k}\left(t\right)\cdot E\left(t\right)\geq f^{\prime }\left(\psi_{\kappa }\left(t\right)\right){\dot{\psi }}_{k}\left(t\right)=\dot{\psi }\left(t\right).$$

More generally, let ${\hat{J}}_{i}$ denote the linear models
$$J_{i}\left(t\right)={\hat{J}}_{i}\left(A_{r}^{t}\right):=\int_{0}^{\infty }\kappa_{i}\left(s\right)A_{r}^{t}\left(s\right)ds,\;i=1,...,n,$$
whose kernels $\kappa_{i}$ are compatible with thermodynamics in the sense that they obey (Equation 104), and let $\psi_{i}\left(t\right)$ be any free energy associated with the *i*-th model satisfying (Equation 105). Thus, we have
$$\psi_{i}\left(t\right)\geq 0\;\mathrm{and}\;{\dot{\psi }}_{i}\left(t\right)\leq J_{i}\left(t\right)\cdot E\left(t\right).$$

For any given pair of kernels $\kappa_{i}$ and $\kappa_{j}$, we can construct a nonlinear current functional and a free energy functional of the form
$$J\left(t\right)=J_{i}\left(t\right)+\alpha \psi_{j}^{\alpha -1}\left(t\right)J_{j}\left(t\right),\;\psi \left(t\right)=\psi_{i}\left(t\right)+{\left(\psi_{j}\left(t\right)\right)}^{\alpha },\;\alpha >1.$$

### 6.2. Integral Representations of the Current

The goal of this subsection is to establish a connection between the nonlinear constitutive functionals proposed in Section 6.1 and constitutive functionals in the form of single, double and triple integrals of the kind proposed by Graffi in [44,45] (see also [7]).

Following the suggestions of Graffi’s paper, we assume a nonlinear constitutive equation of the following general form
$$\begin{array}{cc} J(t) & =\int_{0}^{\infty }S\left(s\right)E^{t}\left(s\right)ds+\int_{0}^{\infty }\int_{0}^{\infty }S\left(s_{1},s_{2}\right)\;E^{t}\left(s_{1}\right)⊗E^{t}\left(s_{2}\right)\;\;ds_{1}\;ds_{2}\\ \; & \;+\int_{0}^{\infty }\int_{0}^{\infty }\int_{0}^{\infty }S\;\left(s_{1},s_{2},s_{3}\right)\;E^{t}\left(s_{1}\right)⊗E^{t}\left(s_{2}\right)⊗E^{t}\left(s_{3}\right)\;ds_{1}\;ds_{2}\;ds_{3}, \end{array}$$
where ***S***, S and S are a second, third and fourth-order tensor-valued function, respectively. A somewhat similar expansion was used in (Equation 59). For arbitrary vectors A,B,C∈V, we have
$${\left(SA\right)}_{i}=\sigma_{ij}A_{j}\;,\;{\left(SA⊗B\right)}_{i}=S_{ijk}A_{j}B_{k}\;,\;{\left(S\;A⊗B⊗C\right)}_{i}=S_{ijkl}A_{j}B_{k}C_{l}\;.$$

The quantities S and S can be taken to be invariant under any permutation of the arguments provided the same permutation is applied to the subscripts jk and jkl.

For simplicity, let us assume that the constitutive relation for J is isotropic. It follows that S vanishes, and ***S***, S must be isotropic tensors. Thus,
$$\sigma_{ij}=\sigma \;\delta_{ij}\;,\;S_{ijkl}=S_{1}\;\delta_{ij}\delta_{kl}+S_{2}\;\delta_{ik}\delta_{jl}+S_{3}\;\delta_{il}\delta_{jk},$$
where $\sigma ,S_{1},S_{2},S_{3}$ are scalar functions of the elapsed time *s*. This implies SA=σA and
$$S\;A⊗B⊗C=S_{1}\;\left(B\cdot C\right)A+S_{2}\;\left(A\cdot C\right)B+S_{3}\;\left(A\cdot B\right)C,$$
so that
(108)$$\begin{array}{cc} J(t) & =\int_{0}^{\infty }\sigma \left(s\right)E^{t}\left(s\right)ds\\ \; & \;+\int_{0}^{\infty }\int_{0}^{\infty }\int_{0}^{\infty }S_{1}\left(s_{1},s_{2},s_{3}\right)\;\left[E^{t}\left(s_{2}\right)\cdot E^{t}\left(s_{3}\right)\right]\;E^{t}\left(s_{1}\right)\;ds_{1}\;ds_{2}\;ds_{3}\\ \; & \;+\int_{0}^{\infty }\int_{0}^{\infty }\int_{0}^{\infty }S_{2}\left(s_{1},s_{2},s_{3}\right)\;\left[E^{t}\left(s_{3}\right)\cdot E^{t}\left(s_{1}\right)\right]\;E^{t}\left(s_{2}\right)\;ds_{1}\;ds_{2}\;ds_{3}\\ \; & \;+\int_{0}^{\infty }\int_{0}^{\infty }\int_{0}^{\infty }S_{3}\left(s_{1},s_{2},s_{3}\right)\;\left[E^{t}\left(s_{1}\right)\cdot E^{t}\left(s_{2}\right)\right]\;E^{t}\left(s_{3}\right)\;ds_{1}\;ds_{2}\;ds_{3}\;. \end{array}$$

The symmetry properties of S gives relations between $S_{1}$, $S_{2}$, and $S_{3}$:(109)$$\begin{array}{c} S_{1}\left(s_{1},s_{2},s_{3}\right)=S_{1}\left(s_{1},s_{3},s_{2}\right)\\ S_{2}\left(s_{1},s_{2},s_{3}\right)=S_{2}\left(s_{3},s_{2},s_{1}\right)\\ S_{3}\left(s_{1},s_{2},s_{3}\right)=S_{3}\left(s_{2},s_{1},s_{3}\right) \end{array}$$
which are apparent in any case from (Equation 108) and
$$\begin{array}{c} S_{2}\left(s_{1},s_{2},s_{3}\right)=S_{1}\left(s_{2},s_{1},s_{3}\right)\\ S_{3}\left(s_{1},s_{2},s_{3}\right)=S_{1}\left(s_{3},s_{2},s_{1}\right) \end{array}$$
among other similar relations. We finally obtain
(110)$$\begin{array}{cc} J(t) & =\int_{0}^{\infty }\sigma \left(s\right)E^{t}\left(s\right)ds\\ \; & +\int_{0}^{\infty }\int_{0}^{\infty }\int_{0}^{\infty }S\left(s_{1},s_{2},s_{3}\right)\;\left[E^{t}\left(s_{2}\right)\cdot E^{t}\left(s_{3}\right)\right]\;E^{t}\left(s_{1}\right)\;ds_{1}\;ds_{2}\;ds_{3}, \end{array}$$
where
$$S\left(s_{1},s_{2},s_{3}\right)=3S_{1}\left(s_{1},s_{2},s_{3}\right).$$

Because of the first equality in (Equation 109) we have $S\left(s_{1},s_{2},s_{3}\right)=S\left(s_{1},s_{3},s_{2}\right)$. Hence, assuming the factorization
$$S\left(s_{1},s_{2},s_{3}\right)=\lambda \left(s_{1}\right)L\left(s_{2},s_{3}\right),$$
it follows that $L\left(s_{2},s_{3}\right)=L\left(s_{3},s_{2}\right)$. Letting ${I\;L}_{ij}=L\delta_{ij}$ in (Equation 106) we obtain
$$\psi_{L}\left(t\right)=\frac{1}{2}\int_{0}^{\infty }\int_{0}^{\infty }L\left(s_{1},s_{2}\right)E^{t}\left(s_{1}\right)\cdot E^{t}\left(s_{2}\right)\;ds_{1}\;ds_{2},$$
and (Equation 110) becomes
$$J\left(t\right)=J_{\sigma }\left(t\right)+2\psi_{L}\left(t\right)J_{\lambda }\left(t\right),$$
where
$$J_{\sigma }\left(t\right)=\int_{0}^{\infty }\sigma \left(s\right)E^{t}\left(s\right)ds,\;J_{\lambda }\left(t\right)=\int_{0}^{\infty }\lambda \left(s\right)E^{t}\left(s\right)ds.$$

Finally, letting λ(s)=L(0,s), we obtain the corresponding free energy
$$\psi \left(t\right)=\psi_{\sigma }\left(t\right)+{\left(\psi_{L}\left(t\right)\right)}^{2}.$$

It is worth noting that this choice of ψ is not unique, but it is the simplest one. Indeed, in general for any pair $L_{1}$, $L_{2}$ such that $L_{1}\left(0,s\right)=L_{2}\left(0,s\right)=\lambda \left(s\right)$, we have
$$\psi \left(t\right)=\psi_{\sigma }\left(t\right)+\psi_{L_{1}}\left(t\right)\psi_{L_{2}}\left(t\right)\;.$$

For the static history $E^{†}\left(s\right)=E\left(t\right)$, $s\in R^{+}$, we have
$$\begin{array}{cc} \psi_{L}\left(t\right) & =\frac{1}{2}\ell \;{\left|E\left(t\right)\right|}^{2},\;\;\;\;\;\;\;\;\ell =\int_{0}^{\infty }\int_{0}^{\infty }L\left(s_{1},s_{2}\right)\;ds_{1}\;ds_{2}>0,\\ J(t) & =\sigma_{0}E\left(t\right)+\lambda_{0}\;\ell \;{\left|E\left(t\right)\right|}^{2}\;E\left(t\right). \end{array}$$
where $\lambda_{0}$ and $\sigma_{0}$ are defined by
$$\lambda_{0}=\int_{0}^{\infty }\lambda \left(s\right)ds,\;\sigma_{0}=\int_{0}^{\infty }\sigma \left(s\right)ds.$$

In particular, by applying the Graffi free energy functional, $\psi_{L}=\psi_{\lambda }^{G}$, we have
$$\ell =2\int_{0}^{\infty }s\lambda \left(s\right)ds.$$

If we restrict our attention to the one-dimensional case, the nonlinear current response J(t) to the application of the static E(t)-valued history is plotted in Figure 2.

## 7. Some Applications to Nonlinear Evolution Problems

The vector space V is now taken to be $R^{3}$. In addition, kernels take values in $Lin\left(V\right)=Lin\left(R^{3}\right)$, which is the space of second-order tensors. In this section, we consider some nonlinear evolution problems arising from the coupling of hereditary models for the electric current with Maxwell equations. This kind of problem describes electromagnetic phenomena in the ionosphere and, more generally, in a nonlinear model of plasma (see [36,46,47], for instance). The corresponding linear model has been scrutinized in [48,49] and extended to explain electromagnetic behavior in a conducting (or imperfect) dielectric such as water (see [36,43,50], for example).

### 7.1. An Energy Inequality for a Nonlinear Plasma

Let $B\subset R^{3}$ be a bounded region occupied by a plasma. According to the Minkowski approach, Maxwell’s equations take the same form in the whole space $R^{3}$, namely
(111)$$\left\{\begin{array}{c} \partial_{t}B+\nabla \times E=0,\\ \partial_{t}D-\nabla \times H+J=0,\\ \nabla \cdot B=0\\ \nabla \cdot D=0. \end{array}\right.$$

Hereafter, $\nabla :=\partial_{x}$ denotes the gradient.

Let H denote the magnetic field. The magnetic induction B and displacement vector D are given by
(112)B=μH,D=εEinB,
where ε and μ are positive constants which stand for the electric permittivity and magnetic permeability of the material, respectively. As usual,
$$\varepsilon =\varepsilon_{0}\left(1+\chi_{e}\right)$$
where $\varepsilon_{0}$ is the vacuum permittivity and $\chi_{e}$ is the electric susceptibility of the material. Thus, we can write $D=\varepsilon_{0}E+P$ by introducing the polarization $P=\varepsilon_{0}\chi_{e}E$. Similarly,
$$\mu =\mu_{0}\left(1+\chi_{m}\right),$$
where $\mu_{0}$ is the vacuum permeability and $1+\chi_{m}$ is the relative permeability of the material so that, after introducing the magnetization $M=\chi_{m}H$, we can write $B=\mu_{0}\left(H+M\right)$.

We assume that the constitutive equation of the electric current J of the plasma is given by
$$J\left(t\right)=\tilde{J}\left(E^{t}\right),$$
where $\tilde{J}$ is a (linear or nonlinear) functional of the electric field history. Let ψ be a free energy functional satisfying (Equation 105).

We first establish a local energy inequality for the plasma evolution system. Multiplying the first equation of the system (Equation 111) by H and the second by E, we obtain by virtue of (Equation 112) the local energy balance
$$\frac{1}{2}\frac{d}{dt}\left(\varepsilon {\left|E\right|}^{2}+\mu {\left|H\right|}^{2}\right)+\nabla \cdot \left(E\times H\right)+J\cdot E=0.$$

From the last inequality in (Equation 105) it follows that
(113)$$\frac{1}{2}\frac{d}{dt}\left(\varepsilon {\left|E\right|}^{2}+\mu {\left|H\right|}^{2}+2\psi \right)+\nabla \cdot \left(E\times H\right)\leq 0.$$

According to [51], we assume the natural boundary conditions at a free plasma surface,
(114)E×n=0,H·n=J·n=0on∂B.

Initial conditions, including the past history of the electric field $E^{0}$, must also be assigned;
(115)$$E\left(0\right)=E_{0},\;H\left(0\right)=H_{0},\;E^{0}\left(s\right)={\hat{E}}^{0}\left(s\right),\;s>0.$$

Now, integrating (Equation 113) over B, it follows
$$\frac{1}{2}\frac{d}{dt}\left(\varepsilon {∥E∥}^{2}+\mu {∥H∥}^{2}+\Psi \right)\leq 0,\;\Psi =2\int_{B}\psi \;dx,$$

∥·∥ being the norm in $L^{2}\left(B\right)$. This is due to the well-known property of the Poynting vector,
$$\int_{B}\nabla \cdot \left(E\times H\right)\;dx=\int_{\partial B}(E\times H)\cdot n\;da=-\int_{\partial B}(E\times n)\cdot H\;da,$$
along with the first boundary condition in (Equation 114). Accordingly, the $L^{2}$ norms of E(t), H(t) and the function $\Psi \left(t\right)=\hat{\Psi }\left(E^{t}\right)$ are bounded for all *t* by their initial values,
(116)$$\varepsilon {∥E(t)∥}^{2}+\mu {∥H(t)∥}^{2}+\hat{\Psi }\left(E^{t}\right)\leq \varepsilon {∥E_{0}∥}^{2}+\mu {∥H_{0}∥}^{2}+\hat{\Psi }\left({\hat{E}}^{0}\right).$$

### 7.2. Boundedness of the Electric Current

By exploiting this energy estimate, we establish here some results about the boundedness of the electric current functional. It is noteworthy that linear and nonlinear models yield different consequences on the $L^{2}$ norm of the electric current.

Let $\hat{J}$ be a linear isotropic functional, which is given by
$$\hat{J}\left(A_{r}^{t}\right)=\int_{0}^{\infty }\kappa \left(s\right)A_{r}^{t}\left(s\right)ds=\int_{0}^{\infty }\sigma \left(s\right)E^{t}\left(s\right)ds,\;\kappa =-\sigma^{\prime },$$
and ψ is a related quadratic free energy, so that its integral over B is equivalent to a weighted $L^{2}$-norm. For instance, when κ>0, $\kappa^{\prime }<0$ and the Graffi functional $\psi_{G}$ is the chosen free energy, then
$$\Psi \left(t\right)=2\int_{B}\psi_{G}\left(t\right)\;dx=\int_{0}^{\infty }\kappa \left(s\right)∥A_{r}^{t}{\left(s\right)∥}^{2}ds={∥A_{r}^{t}∥}_{H_{\kappa }}^{2},$$
where $H_{\kappa }$ denotes the weighted Hilbert space $L_{\kappa }^{2}\left(R^{+};L^{2}\left(B\right)\right)$. Moreover, letting $\kappa_{0}=\int_{0}^{\infty }\kappa \left(s\right)ds$, we have
$$|\hat{J}\left(A_{r}^{t}\right)|\leq \int_{0}^{\infty }\kappa^{1/2}\left(s\right)\kappa^{1/2}\left(s\right)\left|A_{r}^{t}\left(s\right)\right|ds\leq \kappa_{0}^{1/2}{\left[\int_{0}^{\infty }\kappa \left(s\right){\left|A_{r}^{t}\left(s\right)\right|}^{2}ds\right]}^{1/2}$$
and then, the $L^{2}$ norm of the electric current is bounded for all t>0 as well as Ψ,
$${∥J\left(t\right)∥}^{2}\leq \kappa_{0}{∥A_{r}^{t}∥}_{H_{\kappa }}^{2}=\kappa_{0}\Psi \left(t\right).$$

On the other hand, let $\hat{J}$ be a nonlinear functional, for instance
(117)$$\hat{J}\left(A_{r}^{t}\right)=\left[1+\alpha \psi_{G}^{\alpha -1}\left(A_{r}^{t}\right)\right]\int_{0}^{\infty }\kappa \left(s\right)A_{r}^{t}\left(s\right)ds,\;\alpha >1/2.$$

We are forced to restrict our analysis to α>1/2 to prevent the electric current J(t) from assuming singular or non-zero constant values when the null constant history of the electric field $E^{t}=0^{†}$ is considered. Accordingly,
(118)$$\begin{array}{cc} \Psi (t) & =2\int_{B}\left(\psi_{G}\left(t\right)+{\left[\psi_{G}\left(t\right)\right]}^{\alpha }\right)\;dx\\ & =∥A_{r}^{t}{∥}_{H_{\kappa }}^{2}+\frac{1}{2^{\alpha -1}}\int_{B}{\left[\int_{0}^{\infty }\kappa \left(s\right){\left|A_{r}^{t}\left(s\right)\right|}^{2}ds\right]}^{\alpha }dx \end{array}$$

Moreover, applying Hölder’s inequality
$$\begin{array}{cc} |\hat{J}\left(A_{r}^{t}\right)| & \leq \left[1+\alpha {\left(\frac{1}{2}\int_{0}^{\infty }\kappa \left(s\right){\left|A_{r}^{t}\left(s\right)\right|}^{2}ds\right)}^{\alpha -1}\right]\kappa_{0}^{1/2}{\left[\int_{0}^{\infty }\kappa \left(s\right){\left|A_{r}^{t}\left(s\right)\right|}^{2}ds\right]}^{1/2}\\ & \leq \kappa_{0}^{1/2}\left[{\left(\int_{0}^{\infty }\kappa \left(s\right){\left|A_{r}^{t}\left(s\right)\right|}^{2}ds\right)}^{1/2}+\frac{\alpha }{2^{\alpha -1}}{\left(\int_{0}^{\infty }\kappa \left(s\right){\left|A_{r}^{t}\left(s\right)\right|}^{2}ds\right)}^{\alpha -1/2}\right], \end{array}$$
and then, we have the following estimate of the $L^{2}$ norm of the electric current,
(119)$$∥\hat{J}\left(A_{r}^{t}\right){∥}^{2}\leq 2\kappa_{0}\left[∥A_{r}^{t}{∥}_{H_{\kappa }}^{2}+\frac{\alpha^{2}}{4^{\alpha -1}}\int_{B}{\left[\int_{0}^{\infty }\kappa \left(s\right){\left|A_{r}^{t}\left(s\right)\right|}^{2}ds\right]}^{2\alpha -1}dx\right].$$

This allows us to establish the following result.

**Theorem** **1.***Let* (Equation 112) *and* (Equation 117) *be constitutive relations for*
B,D
*and*
J. *In addition, let the initial data be such that*
$∥E_{0}∥,∥H_{0}∥$
*and*
$\Psi \left(0\right)=\hat{\Psi }\left({\hat{E}}^{0}\right)$
*are bounded. Furthermore, we assume that Maxwell’s equations* (Equation 111) *with boundary condition* (Equation 114) *admit solutions. Then, for all*
t>0,
(*i*)
*∥E(t)∥,∥H(t)∥ and Ψ(t) are bounded;*
(*ii*)
*∥J(t)∥ is bounded provided that 1/2<α≤1; otherwise, J(t) is bounded in the $L^{\beta }\left(B\right)$-norm with β=α/(α−1/2).*



**Proof.** Item (i) follows from the energy inequality (Equation 116). In order to establish item (ii), we first assume 1/2<α≤1. By virtue of (Equation 119), there is a positive constant $c_{\alpha }$ such that
$$∥\hat{J}\left(A_{r}^{t}\right){∥}^{2}\leq c_{\alpha }\Psi \left(t\right).$$
applying first the Hölder inequality and then the generalized Young inequality to the last term of (Equation 118), we obtain
$$∥A_{r}^{t}{∥}_{H_{\kappa }}^{2}\leq \Psi \left(t\right)\leq ∥A_{r}^{t}{∥}_{H_{\kappa }}^{2}+{\left|B\right|}^{1-\alpha /2}2^{\alpha -1}∥A_{r}^{t}{∥}_{H_{\kappa }}^{\alpha }\leq 1+c_{0}(\alpha ,|B|)∥A_{r}^{t}{∥}_{H_{\kappa }}^{2},$$
where |B| denotes the Lebesgue measure of B and $c_{0}$ stands for a positive constant dependent on α and |B|. These estimates are similar to those obtained in the linear case. More generally, when α>1/2, we put
β=α/(α−1/2)
and applying the Minkowski inequality ${\left|1+f\right|}^{\beta }\leq 2^{\beta -1}{\left(1+|f\right|}^{\beta })$, β>1, we obtain
$$\begin{array}{c} |\hat{J}\left(A_{r}^{t}\right){|}^{\beta }\leq \kappa_{0}^{\beta /2}{\left[1+\alpha {\left(\frac{1}{2}\int_{0}^{\infty }\kappa \left(s\right){\left|A_{r}^{t}\left(s\right)\right|}^{2}ds\right)}^{\alpha -1}\right]}^{\beta }{\left[\int_{0}^{\infty }\kappa \left(s\right){\left|A_{r}^{t}\left(s\right)\right|}^{2}ds\right]}^{\beta /2}\\ \leq 2^{\beta -1}\kappa_{0}^{\beta /2}\left[{\left(\int_{0}^{\infty }\kappa \left(s\right){\left|A_{r}^{t}\left(s\right)\right|}^{2}ds\right)}^{\beta /2}+\frac{\alpha^{\beta }}{2^{\beta (\alpha -1)}}{\left(\int_{0}^{\infty }\kappa \left(s\right){\left|A_{r}^{t}\left(s\right)\right|}^{2}ds\right)}^{\alpha }\right]. \end{array}$$
hence, there exists some positive constant $C_{\alpha }$ such that
$$∥\hat{J}\left(A_{r}^{t}\right){∥}_{L^{\beta }\left(B\right)}^{\beta }\leq C_{\alpha }\left[∥A_{r}^{t}{∥}_{H_{\kappa }}^{\beta }+\Psi \left(t\right)\right].$$Moreover, if α>1, then β<2 and the generalized Young inequality gives
$$∥\hat{J}\left(A_{r}^{t}\right){∥}_{L^{\beta }\left(B\right)}^{\beta }\leq C_{\alpha }\left[\frac{\alpha -1}{2\alpha -1}+\frac{\alpha }{2\alpha -1}{∥A_{r}^{t}∥}_{H_{\kappa }}^{2}+\Psi \left(t\right)\right]\leq C_{\alpha }\left[1+2\Psi (t)\right].$$
in particular, when α=2, we have β=4/3.    □

## Figures and Tables

**Figure 1 materials-15-06804-f001:**
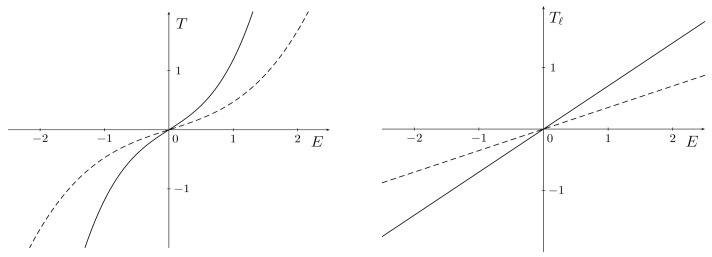
Asymptotic responses in the (E,T)-plane as c→0 (dashed) and c→+∞ (solid), both for linear (on the right) and nonlinear (on the left) constitutive relations with $G_{0}=0.7$ and $G_{\infty }=0.35$.

**Figure 2 materials-15-06804-f002:**
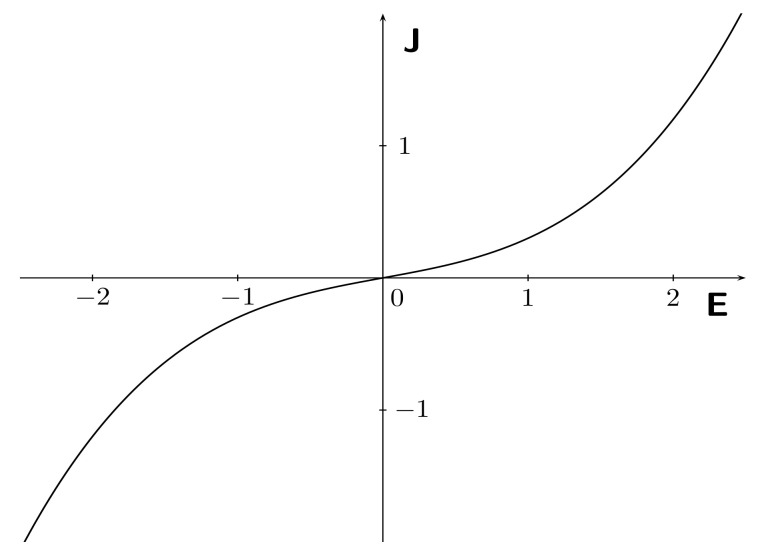
Nonlinear current responses J(t) at different constant values E(t) of the static history $E^{†}$ with $\sigma_{0}=0.2$ and $\lambda_{0}\;\ell =0.1$.

## Data Availability

Not applicable.

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
