# Peer review of "Viscoelastic and Electromagnetic Materials with Nonlinear Memory"

_materials, 2022, doi:10.3390/ma15196804_

Round 1
Reviewer 1 Report
The free energy relating to nonlinear constitutive equations with memory is developed. I think this manuscript is deep in theory, but it seems not outstanding in engineering. Hence it is suggested that the authors deepen the role of engineering in the study. In addition, for the engineering research and practical application of nonlinear and viscoelastic materials, the authors can refer to the following literature: Nonlinear vibrations and stability in parametric resonance of axially moving viscoelastic piezoelectric nanoplate subject to thermo electro mechanical forces
Reviewer 2 Report
Please see the attached file

Reviewer 3 Report
The manuscript presents a method to generate free energies relating to nonlinear constitutive equations with memory. Overall the contribution is well structured, brilliantly written and scientifically sound.
Only a few (very) minor remarks might in my opinion increase the quality of this work:
- The introduction is held very brief. An extension of the paragraph on the materials under consideration (heat/electrical conductors with memory) could increase the appeal for the application oriented reader.
- In Equation 2.22 it is stated that the sum of the linear parameters is equal to 1 for the linear case of 2.20. This conclusion is not comprehensible to me but seems to depend on the relation between \Psi and \Psi_i. Could you please elaborate?
- I am personally not used to the way that the tensor multiplication is handled here. For example in 3.1 \Lambda(s_1) and K(s_1,s_2) are multiplied by \cdot. From my experience, this refers to a single dot operation. However, the second multiplication with \Lambda(s_2) does not have a multiplication sign (which I usually associate with the multiplication of a scalar and a tensor). Similarly, this appears in the line after equation 3.17 where the tensorial product \otimes is introduced. Here I assume the \cdot operation between K and [\lambda\otimes\lambda] refers to a double dot operation. On the other hand, the multiplications in chapter 6.2 on page 20 are well defined in the line before equation 6.9. A similar explanation could potentially help the reader.
- In Equation 7.2 \mu and \varepsilon are used but not properly introduced (I assume these are the magnetic permeability and electric permitivity). Furthermore, as I am not familiar with these formulations for plasma if there might be polarization terms (as in D=\varepsilon E + P)? A short comment could clear this up.
- In Equation 7.4 only the curl free condition is expressed. Could you please elaborate, why there are no other boundary conditions (for example on the normal contribution of the electric displacement or similarly for the magnetic field)
